# Optimised weight programming for analogue memory-based deep neural networks

Charles Mackin [1✉], Malte J. Rasch [2], An Chen[1], Jonathan Timcheck [3], Robert L. Bruce[2], Ning Li [2], Pritish Narayanan[1], Stefano Ambrogio [1], Manuel Le Gallo [4], S. R. Nandakumar [4], Andrea Fasoli[1], Jose Luquin[1], Alexander Friz[1], Abu Sebastian [4], Hsinyu Tsai[1] & Geoffrey W. Burr [1]

Analogue memory-based deep neural networks provide energy-efficiency and per-area throughput gains relative to state-of-the-art digital counterparts such as graphics processing units. Recent advances focus largely on hardware-aware algorithmic training and improvements to circuits, architectures, and memory devices. Optimal translation of software-trained weights into analogue hardware weights—given the plethora of complex memory non-idealities—represents an equally important task. We report a generalised computational framework that automates the crafting of complex weight programming strategies to minimise accuracy degradations during inference, particularly over time. The framework is agnostic to network structure and generalises well across recurrent, convolutional, and transformer neural networks. As a highly flexible numerical heuristic, the approach accommodates arbitrary device-level complexity, making it potentially relevant for a variety of analogue memories. By quantifying the limit of achievable inference accuracy, it also enables analogue memory-based deep neural network accelerators to reach their full inference potential.

[1] IBM Research–Almaden, 650 Harry Road, San Jose, CA, USA. [2] IBM Research–Yorktown Heights, 1101 Kitchawan Road, Yorktown Heights, NY, USA. [3] Stanford University, 450 Serra Mall, Stanford, CA 94305, USA. [4] IBM Research–Zurich, Säumerstrasse 4, 8803 Rüschlikon, Switzerland. ✉email: charles.mackin@ibm.com

The generation, storage, and processing of ever-increasing amounts of data in support of rapid and sophisticated decision-making has spurred remarkable advances in Deep Neural Networks (DNNs) in recent years[1]. DNNs have become ubiquitous within image classification, language processing, prediction, and similar critical tasks across a spectrum of industries. Advancements in deep learning algorithms, architectures, and hardware now enable DNNs to boast near-human—and in some cases—supra-human capabilities. This performance, however, comes at tremendous computational cost in terms of time and energy consumption. A distributed implementation of AlphaGo, which beat the human European champion of the Go strategy board game, required 1,202 CPUs, 176 GPUs, and hundreds of kilowatts[2]. Similarly, a state-of-the-art language prediction model such as Generative Pre-Trained Transformer 3 (GPT-3) contains approximately 175 billion weights, cost tens of millions of dollars to train, and requires approximately eleven Tesla V100 GPUs and thousands of watts for inference[3]. Highly optimised GPUs and tensor processing units (TPUs) form the hardware substrate supporting these systems. Such compute engines, however, are based on conventional von Neumann architectures, in which the memory blocks that store the synaptic weights are physically separate from the computational blocks that process data. This requires high bandwidth and continual data transport between memory and computational blocks, exacting unavoidable time and energy penalties and limiting overall performance (i.e., the 'von Neumann' bottleneck). This has spurred interest in the development of alternative non-von Neumann architectures for DNN acceleration.

DNNs rely extensively on vector-matrix multiplication (VMM) operations, which lend themselves naturally to non-von Neumann crossbar array structures. Within crossbar arrays, analogue memory elements encode the synaptic weights of the network. DNN activations are applied along rows of the memory array, multiplied by the synaptic weights according to Ohm's law, and summed along each column according to Kirchhoff's current law. This enables the crossbar array to implement VMM operations at the location of the data to reduce the impact of the von Neumann bottleneck. This approach was recently shown capable of 280× speedup in per-area throughput while providing 100× enhancement in energy-efficiency over state-of-the-art GPUs[4].

Analogue memory-based DNN accelerators are being widely developed in academia and industry using a variety of memories[5], including resistive RAM (ReRAM)[6,7], conductive-bridging RAM (CBRAM)[8], NOR flash[9–12], magnetic RAM (MRAM), and phase-change memory (PCM)[13,14]. To date, each type of analogue memory exhibits some form of non-ideal behaviour such as limited resistance contrast, significant non-linearity and stochasticity in conductance-vs-pulse characteristics, strong asymmetry during bidirectional programming, read noise, and conductance drift after programming to name a few[15–19]. These memory imperfections ultimately introduce errors into the VMM computations, and can often lead to diminished DNN accuracy relative to state-of-the-art digital systems. That said, state-of-the-art digital systems are currently being optimised to deliver identical DNN accuracies even when activation-precision and weight-precision are reduced from 32-bit floating-point (FP32) to 4-bit integer (INT4) representations or less[20,21]. If DNN models are inherently capable of delivering accurate predictions despite low-digital precision compute, there is a strong expectation that the minimum Signal-to-Noise Ratios (SNRs) within analogue-memory-based systems needed for similar DNN accuracy should not be excessively high.

Incorporating hardware non-idealities within DNN training (i.e., 'hardware-aware' algorithmic training) has been shown effective in making analogue memory-based DNNs more resilient to hardware imperfections[22–25]. Hardware-aware training typically captures various types of memory non-idealities along with circuit nonlinearities such as IR-drops within the crossbar array and activation quantisation due to analogue-to-digital converters (ADCs) and pulse-width modulators (PWMs). Both conventional and novel hardware-aware training produce DNN models comprised of 'unitless' synaptic weights. As shown in Fig. 1, before programming into the analogue memory of choice, these unitless DNN model-weights must be converted into target conductances, typically in units of microSiemens. Since analogue memory weights can be encoded across multiple memory devices, there can be infinitely many ways to implement the same synaptic weight. However, each of these choices for how the weight gets distributed across multiple conductances, will not produce equivalent weight errors[26]. This is further complicated by the fact that DNNs are typically comprised of millions of weights, ranging from large positive to near-zero to large negative weight values. The high degree of inherent interconnectedness present within DNNs also means that any systemic weight errors introduced through sub-optimal weight translation strategies will almost certainly propagate and compound throughout the network. This causes the trained DNN, which has been highly optimised for a specific task, to be perturbed with virtually zero probability of coincidentally landing on a similarly optimal configuration that was not discovered during the training process, especially due to the high dimensionality. This ultimately leads to degraded inference accuracy because there exists a discrepancy between the DNN that was trained—hardware-aware or otherwise—and the analogue memory-based DNN that actually exists in the hardware. Worse yet, in the presence of conductance drift after programming, this degradation is also changing over time.

The remainder of the introduction, along with Fig. 2, introduces several key concepts to further highlight the complexity of the problem being addressed. Analogue memory-based weights typically introduce programming errors due to stochasticity in conductance-vs-pulse curves, device variability, and imperfect yield. An ideal weight programming strategy should determine the target conductances for programming that provide the best-possible outcome—despite errors in programming the conductances at time $t_0$, the subsequent evolution of these weights due to conductance drift, and the read noise associated with performing VMM computations at each point in time (Fig. 2a). Conductance drift is typically modelled using a power law:

$$G(t) = G_0(t/t_0)^{-\nu} \qquad (1)$$

where $G_0$ is the initial conductance at a reference time $t_0$, and $\nu$ is the drift coefficient that determines how the conductance changes with time[27]. Conductance drift is not captured during training, but can be considered during the weight translation process in order to minimise degradations in inference accuracy over time. For instance, if all devices drift with exactly the same $\nu$ coefficient, then we can simply amplify the integrated column currents with a single scaling coefficient that depends only on the elapsed time since programming. The only drawback here being that we might eventually amplify the small amount of background noise such that the overall SNR might start to decrease. Unfortunately, conductances typically have complex drift characteristics where $\nu$ coefficients exhibit stochastic intra-device ('shot-to-shot') variability. Thus we cannot precisely know the value of $\nu$ that will ensue after any given programming event, even for devices that have been carefully characterised. Furthermore, conductances also tend to drift more quickly or slowly depending on the magnitude of the conductance programmed, and the variability in $\nu$ coefficients is sometimes observed to be conductance-dependent as well[28].

In Fig. 2c, each synaptic weight is comprised of multiple conductances with opposing polarities that will drift at different

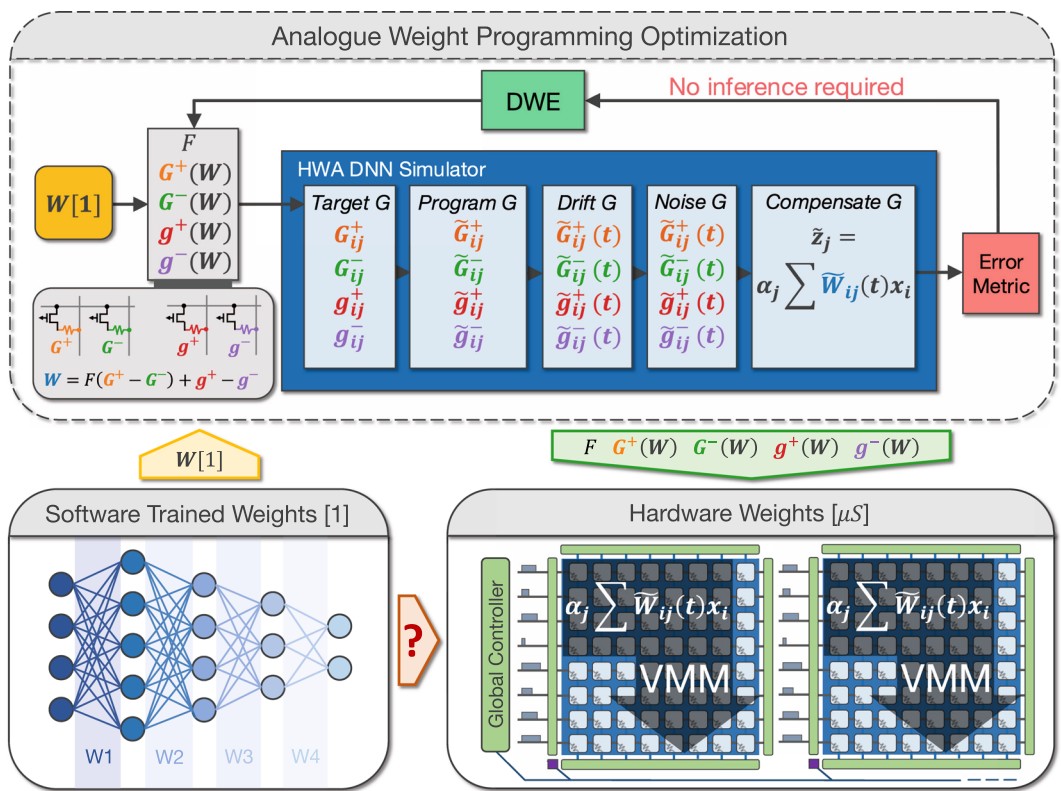

**Fig. 1 Overview of the weight programming optimisation framework.** 'Unitless' weights from software DNN models must be re-scaled into an optimal hardware range (microSiemens), and can be encoded across multiple analogue memory devices: $G^+$, $G^-$, $g^+$, and $g^-$ with varying significance defined by $F$. A weight programming optimisation framework captures all memory imperfections and hardware compensatory techniques, and produces optimal weight programming strategies using an iterative Differential Weight Evolution (DWE) technique to minimise inference accuracy degradations for analogue memory-based DNNs, including degradations that change over time. This can be achieved without the need to run costly inference simulations at multiple time-steps using large datasets.

rates (Fig. 2d) to define the overall evolution of the weight with time (Fig. 2b centre). Conductances within a weight are identically fabricated but may also have varying significance as determined by the scaling factor $F$ between the Most Significant Pair (MSP) and Least Significant Pair (LSP)[29]. This can be advantageous because the MSP can be used to increase the overall dynamic range and program the bulk of the weight, whereas the LSP can be used to fine tune the programmed weight for better precision. The significance factor can be implemented in a number of ways, but is limited to discrete values in this case, which can be readily implemented by multiplying the pulse durations of the input activations applied to the MSP relative to the LSP[30,31]. The use of multiple conductances per weight also introduces a level of redundancy to mitigate device variability and occasional device failures (i.e., imperfect yield).

Figure 2b depicts an example of what is termed a naive programming strategy, in which the majority of the weight is programmed in the MSP, and the LSP is then intuitively used to fine tune the weight in an attempt to eliminate weight errors. The weight distribution being programmed is shown in the background (blue). Programming and drift characteristics are plotted for the individual conductances (Fig. 2d) and the resulting weights (Fig. 2b). The evolution of weight errors $W_E = \alpha\widetilde{W} - W$ as a function of time is depicted, including read noise. Due to the conductance changes over time (Fig. 2d), drift causes weight magnitudes to decline with time, which causes the activations stemming from VMM computations to also decline and adversely affect inference accuracy. As mentioned earlier, this can be mitigated using a drift compensation technique[32], where

activations are amplified close to their original levels using drift compensation factor $\alpha$, which may or may not be uniform along the column-wise dimension of the crossbar array. Drift compensation factors can be calculated using a calibration technique where one or more randomised input vectors are applied to the crossbar array immediately after weight programming. The resulting output activations are saved either locally or off-chip and can be compared with future applications of the same randomised vectors to determine the appropriate drift compensation factor $\alpha$. For this particular set of drift characteristics, a drift compensation factor $\alpha$ of 1.8 is needed after approximately one month (Fig. 2b, e right side).

Finally, Fig. 2e illustrates how an optimised weight programming strategy can allow drift compensated hardware programmed weights $\alpha\widetilde{W}$ to more closely track ideal weights $W$, including as a function of time. This is indicated by the lower standard deviation in weight errors associated with Fig. 2e relative to Fig. 2b. In this case, weight errors are reduced by approximately 39% at $t_0$ immediately after programming and by approximately 17% at one month. In such optimised weight-programming strategies, individual target conductances exhibit complex dependencies on the input unitless software weights: $G^+(W)$, $G^-(W)$, $g^+(W)$, $g^-(W)$ (left side of Fig. 2e). Allowing for these complex programming schemes provides the flexibility necessary to mitigate equally complex distortions in DNN weights resulting from the combination of multiple analogue memory non-idealities. Weight errors also become amplified by the drift compensation factor $\alpha$, making it critical to find optimal weight programming strategies to limit accuracy degradations

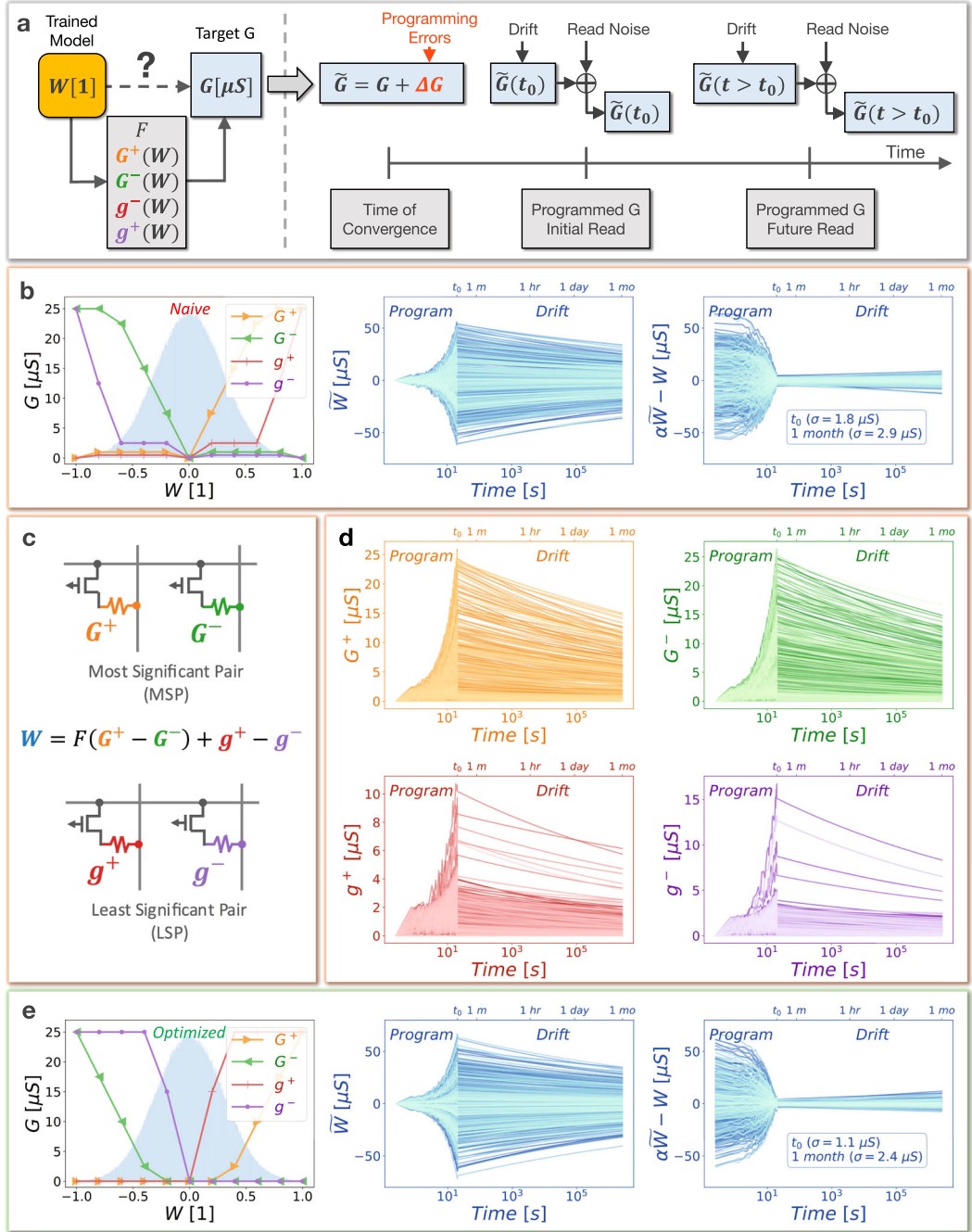

**Fig. 2 Impact of different programming strategies on weight fidelity. a** Unitless weights of software-trained DNN models are translated to analogue memory-based synaptic weights comprised of multiple conductances, which are subject to imperfections including programming errors, conductance drift, and read noise. **b** A sub-optimal weight programming strategy for a distribution of weights (blue) leads to outsized hardware weight errors at $t_0$ that become progressively worse with time, even after drift compensation factor $\alpha$ is applied. **c** Each synaptic weight may be comprised of multiple conductances of varying significance as indicated by the factor $F$, which separates the Most Significant Pair (MSP) and Least Significant Pair (LSP). **d** Individual conductance-programming errors are compounded by subsequent drift over time. **e** An optimised weight programming strategy for the same weight distribution (blue) results in minimal weight errors as indicated visually and by the lower standard deviation in weight errors.

over time. Both examples (Fig. 2b, e) use identical device models for programming errors, drift, and read noise and are re-scaled into the same hardware weight range for fair comparison.

As seen in Fig. 2, given the many different potential error sources injected into VMM computations by analogue memory, along with the complexity of device-level models and the infinite number of potential weight programming strategies, any uninformed or naive weight programming strategy will almost certainly result in sub-optimal weight fidelity and excessive accuracy degradation for analogue memory-based DNNs, particularly as conductances drift over time. This problem is further complicated by the fact that it becomes impractical to run time-consuming inference simulations and evaluate training, validate or test datasets in an iterative fashion—especially at multiple time-steps to include drift—while exploring the vast search-space of possible weight programming strategies.

The overarching objective is to find software-to-hardware translation functions $G^+(W)$, $G^-(W)$, $g^+(W)$, and $g^-(W)$ for weight programming such that

$$\beta_{hw}W = F[G^+(W) - G^-(W)] + g^+(W) - g^-(W) \qquad (2)$$

where $W$ is the unitless software weight, $\beta_{hw}$ is the software-to-hardware weight re-scaling factor, and $F$ is the MSP to LSP significance factor. In this paper, we present a generalised framework capable of producing complex weight programming strategies for analogue memory-based DNNs in light of these constraints. The framework is agnostic to DNN structure, and is shown to generalise well across a variety networks including Long Short-Term Memory (LSTM), Convolutional Neural Networks (CNNs), and Transformers. The numerical framework is capable of accommodating arbitrary device-level complexity and automates the process of finding optimal weight programming strategies—a critical capability given the continual evolution of analogue memory devices. Solving this problem represents a pivotal step towards allowing analogue memory-based DNNs to realise their full energy-efficiency and throughput benefits, while helping to close accuracy gaps with state-of-the-art digital approaches.

## Results

We solve a complex and highly-capable form of weight programming optimisation, in which each synaptic weight is comprised of four conductances $G^+$, $G^-$, $g^+$, and $g^-$ and includes a varying significance factor $F$. This results in a ~$4N$ dimensional parameter space, where $N$ is the number of weights within the network (typically millions). Two additional dimensions are added to the problem: $\beta_{hw}$, which is the scale factor converting the unitless software weights into hardware weights; and $F$, which is the MSP to LSP significance factor. That brings the total number of parameters to $4N + 2$, making the dimensionality of the weight programming optimisation problem potentially larger than the DNN itself. Exploring such a large space, especially with the inference simulations at multiple time-steps within the feedback loop to optimise for drift, quickly becomes intractable. It then becomes critical to reduce the dimensionality of the optimisation problem without hampering the ability to find advantageous weight programming strategies.

Our proposal is to identify the optimal programming strategy for a handful of discretised weights across the useful weight programming range—as opposed to the entire continuous weight distribution—and then linearly interpolate these results for all intermediate weights. The dimensionality can be further reduced by taking advantage of symmetry in DNN weight distributions. Weight distributions for the trained LSTM, CNN, and Transformer networks are highly symmetric about the origin. Thus the programming optimisation results for positive weights, for instance, can be mirrored for negative weights—further reducing the number of parameters by a factor of two. The framework remains readily extensible to strongly asymmetric weight distributions—the dimensionality of the problem simply doubles when it is important to solve for distinct programming strategies for positive and negative weights. For symmetric weight distributions, however, the dimensionality has now been effectively reduced from ~$4N + 2$ to $4D + 2$, where $D$ is the number of discretised positive weights. The reported results make use of six evenly spaced discretised weight intervals. We choose six points as a compromise between capturing the complexity of the optimised weight programming strategy and compute time (as each additional discretised point adds four dimensions to the search space). It is important to note that different memory device characteristics and network weight distributions could lead to scenarios where having more (or fewer) discretisation points is

beneficial. Additional details on how the number of points $D$ impacts the optimised weight programming strategy are provided in the Supplementary Information. As an example, our transformer model, BERT-base, has approximately 86 million weights, of which 53 million are unique. Our dimensional-reduction approach has decreased the number of potential weight programming parameters from approximately 212 million down to twenty-six. Despite this significant reduction, there still exist infinitely many different weight programming strategies to explore, because the search space is still continuous for each conductance parameter and there are infinitely many ways to combine programming strategies for each of the unique discretised weights $D$.

Although the dimensionality of the problem has been reduced to something tractable, it is still important to address the time-consuming inference simulations within the weight programming optimisation loop. Ideally, the best way to gauge the quality of a weight programming strategy is to simulate weight programming using a particular strategy, run inference simulations on the test dataset at multiple time-steps to account for drift, and record the DNN accuracy as a function of time. Because there still exist infinitely many programming strategies to explore, running inference simulations repeatedly at different time-steps to optimise for drift while using large datasets becomes impractical—even given the highly parallelised compute capabilities of a large GPU cluster.

We propose an alternative metric to serve as a proxy for DNN inference accuracy and allow accelerated exploration of the weight programming space, without the need for costly inference simulations within the optimisation loop. We observe that in the limit as weight errors approach zero, hardware weights become exact replicas of the software-trained weights and DNN accuracy becomes identical to the baseline trained accuracy. It then follows that minimising weight errors (i.e., preserving weight fidelity), including across multiple time-steps after conductance programming, should improve DNN inference accuracy over time. One reason this works well is that it remains highly unlikely that introducing large stochastic weight errors will coincidentally move the DNN into a better weight configuration than the one discovered during training, especially given the high dimensionality of the DNN. As a result, the closer a system with imperfect conductances can stay to the initial target DNN weights, the better it should perform.

We propose a time-averaged and normalised mean-squared-error metric as a less computationally expensive proxy for inference accuracy in the weight programming optimisation process. The error metric is

$$\sum_{i=1}^{T} \sum_{j=1}^{D} \kappa_j \sum_{k=1}^{S} \left[ (\alpha_i \widetilde{W}_{ijk}/\beta_{hw} - W_{ijk})/\max(|\mathbf{W}|) \right]^2 \qquad (3)$$

where $T$ is the number of time steps over which to optimise inference accuracy, $D$ is the number of discretised weights selected for optimisation, and $S$ is the number of sample weights simulated at each discretised weight to estimate variance in weight errors. $\kappa_j$ is the relative importance of the discretised weight within the DNN weight distribution, $\alpha_i$ is the drift compensation factor, $\widetilde{W}_{ijk}$ is the target weight including all hardware-associated errors (e.g., programming errors, conductance drift, read noise), $\beta_{hw}$ is the software-to-hardware weight distribution re-scaling factor, $W_{ijk}$ is the ideal unitless target weight from software, and $\mathbf{W}$ represents the entirety of the unitless software DNN weight distribution. Minimising mean-squared-error encourages weight errors to be normally distributed with zero mean, which helps prevent introducing unwanted bias terms that would adversely impact accuracy. This error metric is normalised

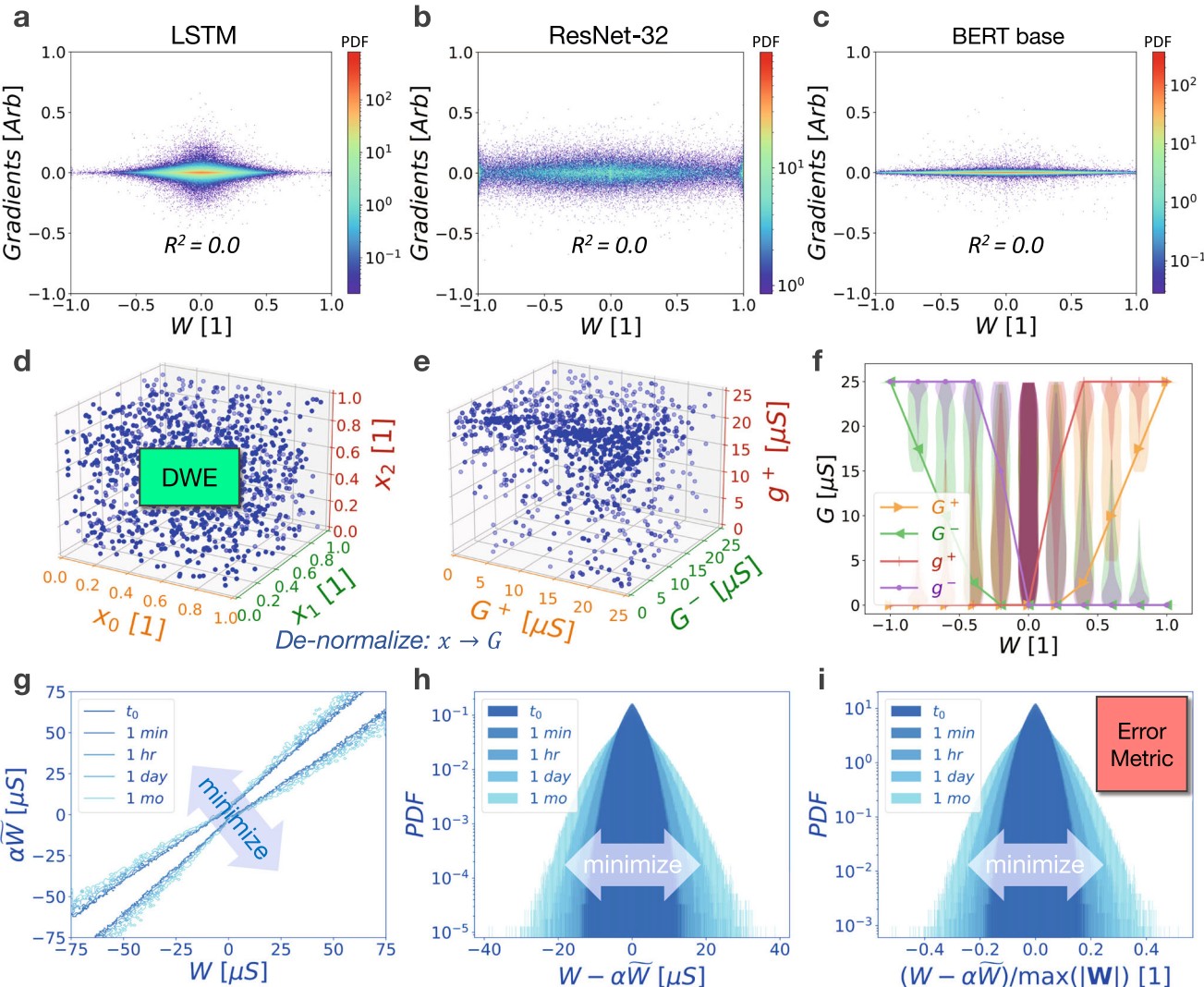

**Fig. 3 Algorithmic details of weight programming optimisation.** DNN gradients are uncorrelated with weight value as shown for (**a**) LSTM, (**b**) ResNet-32, and (**c**) BERT-base. This leads to a weight error importance $\kappa_j$, as defined in our error metric, which depends solely on weight density. The weight programming parameter space is then explored using (**d**) Differential Weight Evolution (DWE) on parameter vectors $x$ within a $\sim 4D + 2$ dimensional hypercube, where $D$ represents the number of positive discretised weights. **e** De-normalised hypercube parameters produce valid conductance combinations that capture optimisation constraints due to conductance inter-dependencies. **f** A two-dimensional projection of the weight programming strategies explored, including the optimal solution (solid lines). Background violin plots show coverage of the weight programming space explored and reveal underlying programming constraints. **g** Outlines of correlation distributions for drift compensated hardware weights $\alpha\widetilde{W}$ versus ideal weights $W$ showing an outward diffusion over time. **h** The corresponding probability density function of weight errors across all weight magnitudes, showing a similar outward diffusion with time. **i** The resulting normalised weight error distribution used to define the error metric.

by the weight range max ($|\mathbf{W}|$) to minimise errors in relation to the overall width of the distribution.

Our error metric includes a temporal component to enable DNN inference optimisation over time in the presence of drift. We opt for a time-averaged error metric that implies all time steps are of equal importance. This is equivalent to saying inference accuracy at one second is just as important as inference accuracy at one hour. This time weighting, however, is easily modified to account for situations where inference accuracy may be non-uniformly important over time. It may also be beneficial to introduce different temporal weighting schemes, or to organise the time-steps in a non-uniform (e.g., logarithmic) way (due to the power-law nature of conductance drift in phase-change memory (PCM), for instance). Lastly, all weight errors are treated as equally important, which results in errors being weighted using $\kappa_j$ according to their relative frequency (density) in the DNN

weight distribution. This stems from the fact that we find zero correlation between weight values and their gradients during the last epoch of training (Fig. 3a–c). These gradients are a direct estimate of the adverse impact of weight perturbations (errors) on the DNN loss function (accuracy) during the last steps of training, and thus can serve as a proxy for the network's sensitivity to errors on each weight.

Figure 3 provides a high-level overview of several key steps in the weight programming optimisation process for three very different DNNs: LSTM, ResNet-32, and BERT-base (Fig. 3a–c). Optimisation is performed on a $4D + 2$ dimensional hypercube (Fig. 3d) because $W = F(G^+ - G^-) + g^+ - g^-$ represents a hyperplane, which reduces the dimensionality of the search space but also adds constraints to the optimisation problem. To avoid $4D$ inter-dependent conductance constraints, optimisation is performed on a hypercube, which is then 'de-normalised' into

valid conductance combinations in an intermediate step shown in Fig. 3e. Further details on this denormalisation process can be found in the Supplementary Information.

Figure 3f represents a two-dimensional projection of the weight programming exploration space, including an example of an optimised weight programming strategy $G^+(W)$, $G^-(W)$, $g^+(W)$, and $g^-(W)$ indicated by solid lines. Violin plots in the background highlight the range of conductance values being explored, illustrating some of the conductance constraints. For instance, there is a great amount of overlap in violin plots at 0 $\mu S$ because small weights can be constructed using small and large conductances alike. Large positive weights, however, must be constructed using large $G^+$ and $g^+$ values while minimising $G^-$ and $g^-$ values, especially if we are to make effective use of the dynamic range and minimise relative weight errors. This is evidenced by the reduction in the range of valid conductance combinations explored when determining how to program larger-magnitude weights.

Figure 3g outlines the correlation between hardware weights and ideal weights at various points in time, but with the drift compensation factor $\alpha$ included, so that correlation plots for different points in time now overlap each other. In order to distinguish the distributions at different times, each has been plotted only at the locus of points corresponding to an outline of the entirety of the distribution (i.e., capturing 100% of weights). Thus optimisation seeks to bring these curves closer to the diagonal, yet because weights drift in complex ways due to conductance-dependent $\nu$ values and $\nu$ variability, the weight errors become progressively worse and the curves inherently move out away from the diagonal over time. The objective here, as described in the error metric, is to preserve weight fidelity to the extent possible given the complex behaviours of the different underlying devices models, and to do so across multiple time steps. The distribution of the weight errors depicted in Fig. 3h shows a similar inevitable outward diffusion over time due to drift. Lastly, Fig. 3i shows the same distribution after normalisation with respect to the maximum weight, which reflects how our error metric is computed. Weight errors depicted in Fig. 3d–f have not been discretised to better conceptually illustrate what is happening to the DNN weight distribution as a whole.

Although the constrained optimisation problem has been transformed into the exploration of a hypercube, simultaneously accommodating multiple complex and stochastic analogue memory device models still translates into optimising a high-dimensional non-convex and stochastic (e.g., noisy) error metric. Numerically evaluating the Hessian of the error metric reveals it is not positive semi-definite. This renders gradient-descent-based optimisation techniques—which could potentially further accelerate the search through the weight programming space—ineffective. We also find that combinations of gradient-descent and basin-hopping (i.e., simulated annealing) fail to reliably find adequate minima. Additional information regarding the difficulty of finding suitable minima is reported in the Supplementary Information. We report, however, that the evolutionary algorithm known as Differential Evolution[33], when well-tuned and populated with different starting points, consistently performed well and identified good weight programming strategies. As a result, we appropriately refer to this heuristic optimisation strategy as Differential Weight Evolution (DWE).

Now that we have enabled extensive optimisation in a high-dimensional space across multiple time-steps—in a way that is completely agnostic to network structure, size, and test datasets thanks to the use of a proxy error metric—the question becomes whether the resulting weight programming strategies can actually materially improve inference accuracy in analogue memory-based DNNs.

**Generalisation across different DNN types.** We simulate the programming of tens of millions of weights according to the strategies derived from the weight programming optimisation computational technique, and evaluate a variety of DNNs and test datasets at multiple time-steps, to show that weight programming optimisation consistently enhances the accuracy of analogue memory-based DNNs. Analogue memory device models are experimentally derived from mushroom-type phase-change memory (PCM) devices. Figure 4a–c depicts the set of stochastic device models that define weight programming errors, drift coefficients, and read noise, all as a function of conductance. For each device characteristic, both the average response (solid line) and the variability around this (shaded region, corresponding to plus-minus one standard deviation) are used by the optimisation algorithm.

We find that our weight programming optimisation generalises well across different DNN types, including recurrent neural networks (RNNs), Convolutional Neural Networks (CNNs), and Transformer-based networks. All reported inference results are produced using simulated hardware as opposed to physical hardware. For RNNs, we evaluate a two-layer Long Short-Term Memory (LSTM) network on the Penn Treebank dataset[34] (Fig. 4d). For CNNs, we examine a ResNet-32 network using the CIFAR-10 dataset[35] (Fig. 4e). And for Transformer-based networks, we evaluate BERT-base on the MNLI dataset[36] (Fig. 4f). In this way, we not only demonstrate that weight programming optimisation enhances the accuracy of analogue memory-based DNNs relative to sub-optimal programming strategies, but we also provide evidence that our optimisation approach generalises well across a wide variety of very different network architectures. These accuracy enhancements are achieved while being completely agnostic to any network feature other than the weight distribution including network type, size, structure, complexity, type of nonlinear activations function employed, etc.

The results for several non-optimised or naive programming strategies are shown in Fig. 4d–f as a reference to clearly demonstrate the added benefit of weight programming optimisation. In the first naive weight programming strategy, the entirety of the weight is programmed into the MSP, while making minimal use of the LSP and using an $F$ factor of one. This represents an intuitive approach in which one simply tries to converge on the target weight as quickly as possible using the MSP and subsequently uses the LSP only for fine-tuning. In the second naive approach, weights are split equally between the MSP and LSP using an $F$ factor of one. Introducing such redundancy has been shown to offer accuracy benefits by effectively countering some of the variability present in analogue memory[26,37,38].

It is important to note, that even before any optimised weight programming is brought to bear, Fig. 4d–f vividly illustrates the benefits of hardware-aware (HWA) training (solid) relative to 32-bit floating-point (FP) trained networks (dashed), even when using naive programming strategies (MSP only, MSP/LSP (50/50)). Simulation results are the compilation of twenty-five independent weight programming and inference simulations over time, showing the average result with central lines surrounded by shaded regions representing plus-minus one standard deviation. By subjecting the DNN to memory and circuit non-idealities during training[22–25], HWA training clearly makes the DNN more resilient to the various hardware non-idealities, and significantly enhances network accuracy relative to floating-point training across the board. HWA training alone, however, is insufficient for achieving and maintaining iso-accuracy as compared to the training baseline, especially as weights evolve after programming due to conductance drift.

Weight programming optimisation is thus introduced to augment HWA training. As shown by the green lines in Fig. 4d±f, the combination of HWA training together with weight programming optimisation further enhances DNN accuracy,

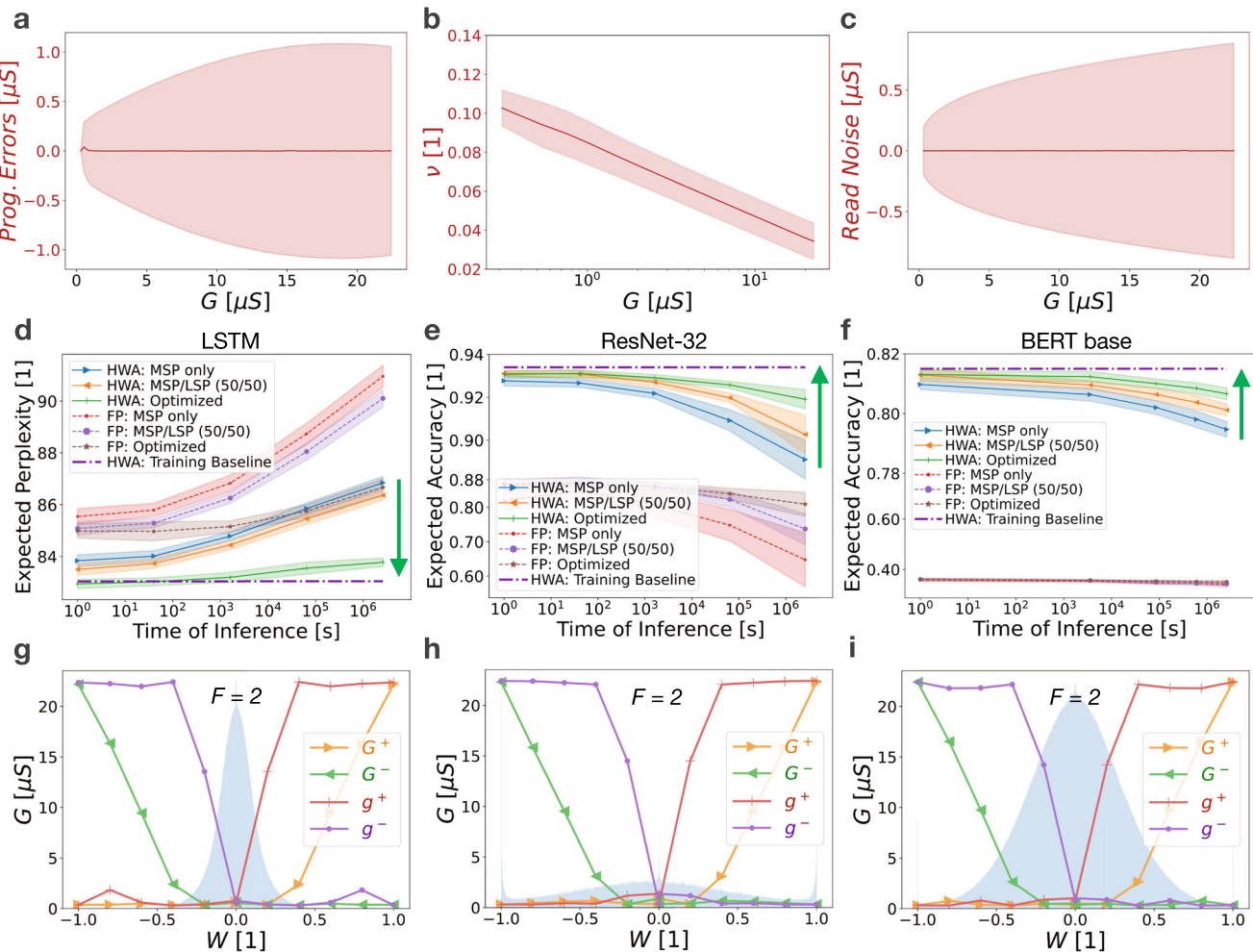

**Fig. 4 Weight programming optimisation improves inference accuracy.** Stochastic analogue memory device models, in this case derived from the measurement of mushroom-type phase-change memory (PCM) devices, for (**a**) conductance-dependent programming errors, (**b**) conductance-dependent drift coefficients, and (**c**) conductance-dependent read noise, with solid red lines representing the mean and shaded red regions representing plus-minus one standard deviation. Simulated inference results show the benefits of both hardware-aware (HWA) training[22] and the weight programming optimisation process introduced in this paper, showing good generalisation across (**d**) a recurrent neural network such as a two-layer Long Short-Term (LSTM) network evaluated on the Penn Treebank dataset, (**e**) Convolutional Neural Networks such as ResNet-32 evaluated on the CIFAR-10 dataset, and (**f**) Transformer-based networks such as BERT-base, evaluated on the MNLI dataset. Average inference performance and plus-minus one standard deviation are denoted by lines and shaded regions, respectively. Target baselines (dash-dot purple) are computed using conventional (i.e., non-hardware-aware) training using 32-bit floating-point (FP) precision. **g–i** The corresponding optimised programming strategies (solid lines) and weight distributions (blue highlight) for each network. Inference simulation results are compiled from twenty-five independent inference accuracy simulations over time for various training and weight programming strategies. The optimal MSP/LSP significance factor $F$, also a parameter solved for in the weight programming optimisation process, was determined to be two in each scenario.

driving the inference accuracy (solid green) as close as possible—given the underlying memory non-idealities—to iso-accuracy with the trained model (dashed-dot purple line). Weight programming optimisation is able to devise much more complex weight programming strategies where programmed conductances can be functions of the unitless weight: $G^+(W)$, $G^-(W)$, $g^+(W)$, and $g^-(W)$. Optimal programming strategies for each DNN are depicted in Fig. 4g–i and are quite similar, which is reasonable given that each DNN is implemented using the same device models. This also demonstrates repeatability of the heuristic given that each optimisation initialised with a completely random set of programming strategies. It is interesting to note that Fig. 4h finds an optimal weight programming strategy that programs non-zero values into the $g^-$ conductance even when the overall weight being implemented is positive. It is precisely these complex and potentially counter-intuitive programming strategies, which also

translate into improved inference accuracy, that our optimisation framework is able to find. Minor variations in programming strategies are likely due to differences in the DNN weight distributions. Hyper-parameter scans for the hardware-aware and 32-bit floating-point training for each DNN are provided in the Supplementary Information.

It is the combination of hardware-aware DNN training and subsequent weight programming optimisation that drives inference accuracy as close as possible to iso-accuracy for analogue memory-based DNNs. As such, weight programming optimisation represents a computational technique that can contribute toward the eventual elimination of accuracy gaps between analogue memory-based DNNs and state-of-the-art digital approaches. This in turn enables analogue memory-based DNNs to better highlight their energy-efficiency and per-area through-put benefits, while minimising potential trade-offs in accuracy.

**Generalisation across different device models.** We now modify the underlying analogue memory device characteristics and repeat the weight programming optimisation, to see if our optimisation approach generalises well to different device models. This is a critical feature given that analogue memories are continually being modified and improved. If our weight programming optimisation technique generalises well across different device characteristics, we can effectively automate the process of finding optimal weight programming strategies. This represents an important step not just for closing potential accuracy gaps, but also for establishing a way to reliably and rapidly connect device characteristics to resulting DNN accuracy. Weight programming optimisation allows one to determine, for the first time to our knowledge, the expected best-case inference accuracy potential for a given set of complex analogue memory characteristics, using a modest set of simulations for each network type. As a result, we can now effectively and objectively compare proposed devices against each other in terms of best-possible DNN inference performance. Weight programming optimisation then becomes a critical tool for guiding the evolution of analogue memory devices.

This is depicted in Fig. 5, where the underlying analogue memory conductance drift model has been modified to a match a different phase-change memory (PCM) device previously reported[28]. Despite different conductance-dependent and stochastic conductance drift models, the weight programming optimisation again effectively drives the inference accuracy results as close to the hardware-aware training baseline as possible (dash-dotted line). Comparison of Figs. 4 and 5 shows that the weight programming optimisation technique generalises well across different device models. This comparison also shows, however, that the analogue memory device of Fig. 5 actually performs worse across the board for LSTM, CNN, and Transformer models relative to the device described in Fig. 4, when both are evaluated in the limit of what is optimally achievable with either device. This is counter-intuitive because the memory device of Fig. 5 provides a larger dynamic range with $g_{max} = 30\mu S$ and exhibits lower conductance-dependent drift on average. Furthermore, if one had compared these two devices under naive programming strategies, one might have incorrectly concluded that the device of Fig. 5 was better (compare the orange curves for HWA: MSP/LSP (50/50) between Figs. 4d and 5d).

Because our computational technique enables the extraction of optimal programming strategies and the corresponding maximum accuracy potential for each set of device characteristics, we can now more definitely say that it is preferable to implement DNNs using the device of Fig. 4 than the device introduced in this section. This is a key finding. Figures 4 and 5 show that there is considerable spread or variability in inference accuracy results when sub-optimal weight programming strategies are employed. In the absence of the weight programming optimisation approach introduced in this paper, this uncertainty makes it very virtually impossible to evaluate—analytically or through intuition—the true inference potential from a given set of device characteristics. Our weight programming optimisation approach can thus—given a fairly modest set of conductance-programming, drift and noise characteristics (Figs. 4a–c and 5a–c)—provide uniquely accurate feedback as to which device will eventually provide the best DNN accuracy.

Interestingly, the derived programming strategies shown in Figs. 4 and 5 are quite similar to each other. This is likely because while the underlying device drift model changed between the two devices, the programming error model—which exerts a large influence on the resulting optimal programming strategy—remained quite similar. We also note

that $F = 2$ produced the optimal weight programming strategy, probably because this choice increases the overall dynamic range of the weight distribution. One might intuitively think this implies one should program the bulk of the weight in the MSP first, and then use the LSP for fine tuning. In contrast, our weight programming optimisation framework chooses to program the entirety of the weight in the LSP whenever possible, and only makes use of the MSP for larger weights when it becomes absolutely necessary. This is because any programming errors in the MSP get amplified by the $F = 2$ factor—the strategy thus avoids such error amplification whenever possible. These types of programming strategies can be counter-intuitive at first glance, but often make sense in hindsight. The beauty of this weight programming optimisation process is that it reliably automates finding these strategies, and does so in a quantitative fashion. A series of LSTM and ResNet-32 weight programming optimisation results are provided in the Supplementary Information across a variety of conductance-drift models. These results provide further evidence that this computational technique can reliably identify optimal programming strategies, and that the resulting inference accuracy consistently outperforms naive and other manually-constructed programming strategies.

## Discussion

Optimal translation of software-trained weights into analogue hardware weights represents a critical step in achieving and maintaining iso-accuracy over time for analogue memory-based DNNs. We report a computational framework that automates the process of crafting complex weight programming strategies for analogue memory-based DNNs in order to minimise accuracy degradations during inference, including over time. We solve a complex and highly flexible form of weight programming optimisation, where each synaptic weight is comprised of four conductances $G^+$, $G^-$, $g^+$, and $g^-$ of varying significance $F$. The optimisation framework is agnostic to all DNN features (e.g., size, layer type, activation function) with the exception of weight distribution, and is shown to consistently improve inference accuracy across a variety of networks including Long Short-Term Memory (LSTM), Transformer, and Convolution Neural Networks (CNNs).

This highly flexible numerical heuristic accommodates arbitrary device-level complexity, making it potentially relevant for a variety of analogue memories with rapidly-evolving device characteristics. Our approach also identifies the limit of achievable inference accuracy given the imperfections in analogue memory. As such, this optimisation framework represents a new and critical tool for enabling analogue memory-based DNNs to reach their full inference potential. This capability also allows analogue memory characteristics to be more objectively compared, since we can now readily evaluate the best-possible accuracy potential of new devices, as constrained by the complex and subtle interplay of their memory non-idealities. Interestingly, this computational technique optimises inference accuracy without ever running inference simulations or evaluating training, validation, or test datasets. It should also be pointed out that weight programming optimisation represents a critical step in translating DNNs into analogue hardware, irrespective of how those DNNs were originally obtained—through hardware-aware training or otherwise. We focus primarily on hardware-aware trained DNNs in this work to demonstrate that weight programming optimisation can extend and augment the benefits of hardware-aware training. These largely independent steps, when combined, help analogue memory-based DNNs reach and maintain inference

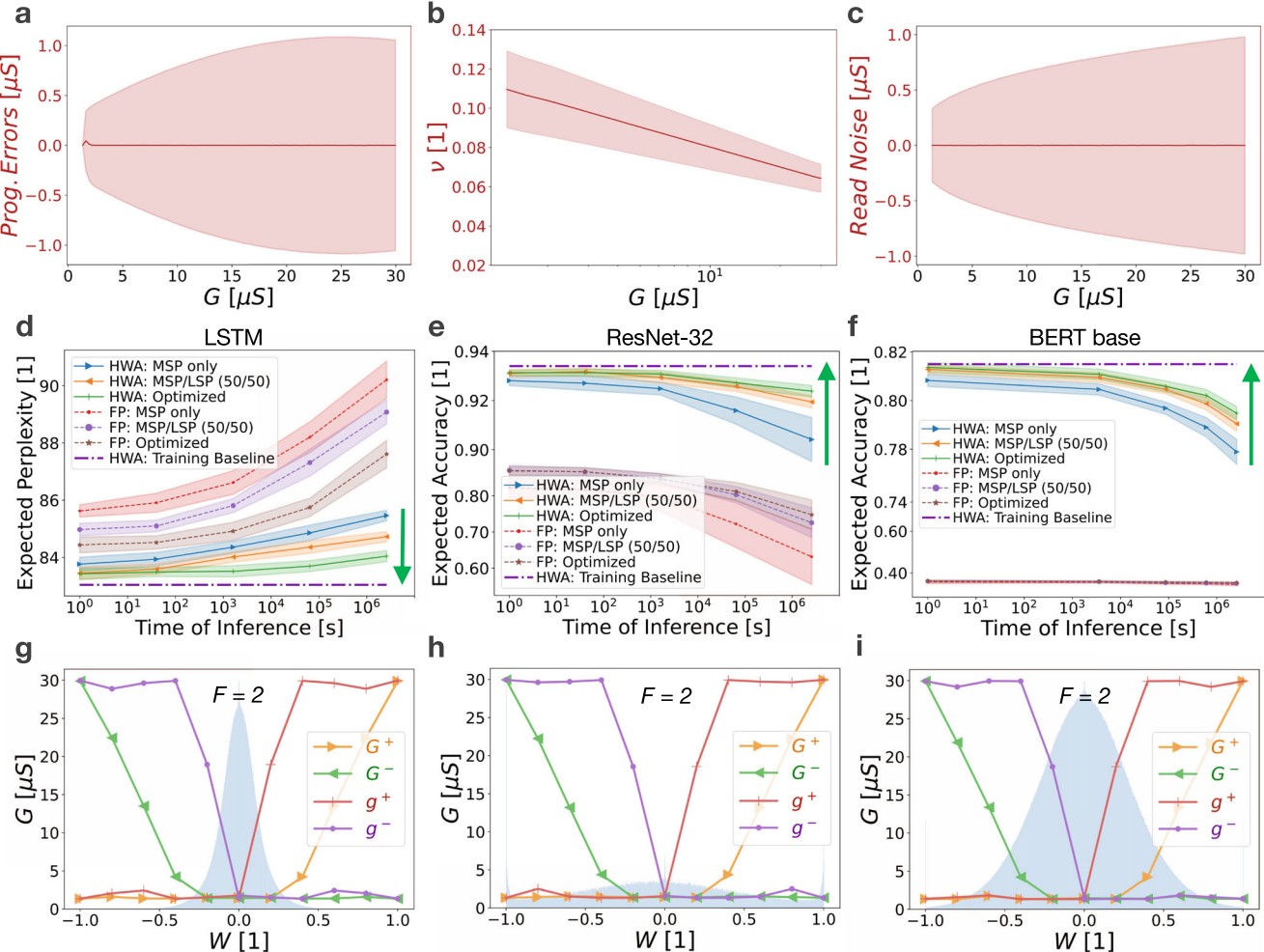

**Fig. 5 Weight programming optimisation improves inference accuracy for a different set of device characteristics.** An alternative device with different underlying stochastic analogue memory device models for (**a**) conductance-dependent programming errors, (**b**) conductance-dependent drift coefficients, and (**c**) conductance-dependent read noise, with solid red lines representing the mean and shaded red regions representing plus-minus one standard deviation. Simulated inference results still generalise well across (**d**) a two-layer Long Short-Term (LSTM) network evaluated on the Penn Treebank dataset, (**e**) ResNet-32 evaluated on the CIFAR-10 dataset, and (**f**) BERT-base evaluated on the MNLI dataset. Although this device exhibits better performance under naive programming strategies (compare orange curves in part (**d**) against Fig. 4d), the best-possible inference performance achievable with this device is worse than the device used for Fig. 4. Average inference performance and plus–minus one standard deviation are denoted by lines and shaded regions, respectively. **g–i** the corresponding optimised programming strategies for each network are similar to those in Fig. 4, with only subtle changes. Simulation results are compiled from twenty-five independent inference accuracy simulations over time for various training and weight programming strategies. The optimal MSP/LSP significance factor $F$ was determined to be two in each scenario.

accuracies that are equivalent (or near equivalent) to 32-bit floating-point trained DNNs.

All weight programming in this work was performed on the entirety of the DNN weight distribution. Weight programming optimisation could, however, be performed individually for each crossbar array within the network. Similarly, drift compensation can be readily performed column-wise in hardware, implying that weight programming optimisation could potentially be performed uniquely for each array-column within the analogue memory. While this would likely lead to additional accuracy improvements, this would not be feasible without considerable numerical acceleration of the presented technique, in order to run what will likely become hundreds of thousands of independent weight programming optimisation simulations in parallel.

It is important to emphasise that the weight programming optimisation presented is not dependent on any unique hardware information and is not a form of calibration. Instead, weight programming optimisation represents a one-time computational

cost that should be performed for each unique DNN and unique set of underlying analogue memory device characteristics. The optimised weight programming strategy can then be used to program all instances of that DNN into devices that exhibit those particular device characteristics. Finally, one can imagine more complex weight programming optimisation frameworks that incorporate additional considerations such as minimisation of energy consumption by the analogue memory. In these cases, our approach could be adapted to include programming strategies that not only drive DNNs towards high inference accuracy, but also consider the implications of different weight implementations on the energy-efficiency of DNNs during inference.

## Methods

**PCM characterisation.** The analogue memory characteristics reported in this work stem from mushroom-type phase-change memory (PCM) devices comprised of doped germanium-antimony-tellurium (GST)[28]. PCM devices are initially

conditioned using $10^5$ full RESET pulses, which melt and rapidly quench the PCM material into an amorphous (i.e., minimum conductance) state. This is achieved using 100 ns pulse durations with amplitudes of 4.5V. A full SET (i.e., maximum conductance) pulse has a voltage amplitude of approximately 2V, a pulse duration of 1 $\mu s$, and a pulse trailing edge of 1 $\mu s$. The programming of the full SET state and intermediate analogue states is achieved through careful optimisation of SET and RESET pulses, which includes exploring various combinations of SET pulse voltages, durations, and trailing edges. Errors in programming various conductance states lead to the conductance-dependent programming error models reported in this work. Conductance drift was measured over a period of 1000 seconds using twenty points per decade. Drift coefficients are obtained by fitting conductance versus time using a power-law dependence. The mean and standard deviation in drift coefficients were extracted for various conductance values to produce the conductance-dependent drift characteristics reported in this work. Read noise was characterised by performing 1000 sequential current measurements using one second time spacing. Read noise is then extracted after subtracting drift from the conductance versus time data. This was performed for a range of conductance values to produce conductance-dependent read noise characteristics. In all measurements, PCM conductances are measured by applying a fixed read voltage of 0.2 V and measuring the resulting current.

**Hardware-aware training**. It has been shown that hardware-aware training in software is crucial for improving accuracy for analogue inference[19,24,25,39]. We follow the spirit of these earlier studies by incorporating a variety of hardware-specific non-idealities during the forward propagation phase of hardware-aware training. In detail, for hardware-aware training, we add weight noise of similar strength and characteristics of the programming noise to each weight matrix during training. Since a new instance of this weight noise is added each mini-batch, it acts as a regularizer during stochastic gradient descent (SGD) and improves noise robustness after convergence. Additionally, we clip and quantise (8-bit) input activations to mimic pulse-width modulators (PWMs), and clip and quantise (10-bit) output activations to mimic analog-to-digital converters (ADCs), and add output noise (Gaussian with a standard deviation on the order of an LSB). Other factors such as ADC nonlinearities are not considered in this work, but can be readily incorporated. We also add cycle-to-cycle (updated each VMM) read noise and modify the output activations to mimic the effects of IR-drop within the crossbar array. Conductance drift is not included because its time-dependence is not readily incorporated into the loss function during training. Similarly, the use of multiple conductances per weight is not included because optimal weight splitting, which also exhibits time-dependence, cannot readily be incorporated into the back-propagation algorithm and loss function during training. All models are trained in normalised weight units by imposing a clipped weight range on the interval $(-1, 1)$ during the hardware-aware training. Our weight programming framework can then be used to optimally convert these abstract trained weights into analogue hardware weights.

The input and output dynamical ranges of the crossbar are fixed during hardware-aware training to the intervals $(-1, 1)$ and $(-10, 10)$, respectively. We also train one additional input scale (for each crossbar) and output scales and offsets (for each column of the crossbar array) to improve the mapping of activation inputs and outputs to crossbar array inputs and outputs. We scan a variety of hyper-parameters during training to arrive at hardware-aware DNNs with accuracies that are equivalent (or near equivalent) to their conventionally trained 32-bit floating-point counterparts. We find the most important hyper-parameter sweeps are the learning rate and strength of the weight noise (further details provided in the Supplementary Information). All hardware-aware training starts from conventionally trained DNNs obtained using standard SGD methods and 32-bit floating-point precision. Inference simulations are evaluated on the test datasets at various time steps and include weight programming errors and conductance drift while retaining the non-idealities of the forward pass previously mentioned such cycle-to-cycle read noise, output noise, and quantisation (and clipping) effects induced by PWMs and ADCs.

**Delayed verification**. This work assumes that the analogue memory devices within crossbar arrays are programmed in a row-wise iterative fashion using a delayed verification strategy. Because analogue memory devices can exhibit some degree of conductance instability after the application of a programming pulse, it makes sense to maximise the time between successive programming pulses to allow the analogue memory devices as much time as possible to stabilise. We therefore cycle through the rows of the crossbar array many times while applying only one programming pulse per row (where appropriate), as opposed to programming an entire row to completion before moving onto the next row. This results in $G(t) = G_0(t/t_0)^{-\nu}$, where $t_0 = 20$ seconds. We have previously employed this weight programming time-scale as an effective compromise between conductance stability and programming speed for the programming of millions of weights[40].

**Differential weight evolution**. We employ the Scipy implementation of differential evolution for its ability to effectively search large non-convex and stochastic candidate spaces. Other gradient descent-based optimisers including standalone or combinations of simulated annealing (i.e., basin-hopping) with local gradient descent-based methods were found to be ineffective at finding advantageous

weight programming strategies. We fine tune a number of parameters to work well across DNN types and varying analogue memory device models. We use a population size of 100 initialised with Latin hypercube sampling. Optimisation is parallelised using the maximum available CPUs (e.g., 'workers') per compute node, which is sixteen in our case. A recombination parameter of 0.6 is used along with dithering parameters of (0.0, 0.2), which change the mutation constant on a generation by generation basis. The termination criterion is a relative tolerance (e.g., 'tol') of 0.05, meaning that the population has converged on a solution with minimal variation. The absolute tolerance (e.g., 'atol') was set to 0.0. We allow for a small errors around weight hyperplanes using a parameter $\Delta G$, which provides added flexibility to slightly 'over-program' or 'under-program' weights in anticipation of non-uniform drift rates to potentially minimise the error metric more effectively. This confinement around the weight hyperplane is defined by $W - \Delta W \leq F(G^+ - G^-) + g^+ - g^- \leq W + \Delta W$, where $\Delta W \approx 2(F + 1)\Delta G$. Details regarding the hypercube denormalisation to capture inter-dependent conductance constraints are included in the Supplementary Information due to space limitations.

## Data availability
The training and test datasets used for this study are publicly available[34–36]. The raw data that support the findings of this study can be made available by the corresponding authors upon request after IBM management approval.

## Code availability
The weight programming optimisation code that supports the findings of this study is available from the corresponding author after IBM management approval on a case-by-case basis. The code for our hardware-aware training and inference simulator cannot be publicly released without IBM management approval and is restricted for export by the US Export Administration Regulations under Export Control Classification Number 3A001.a.9. For similar analogue memory-based DNN implementations and training, we refer the readers to our open source Apache License 2.0 IBM Analog Hardware Acceleration Kit at https://github.com/IBM/aihwkit[25].

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

## Acknowledgements

The authors acknowledge IBM's AI Hardware Center (https://research.ibm.com/collaborate/ai-hardware-center) and management support from W. Wilcke, S. Narayan, J. Welser, J. Burns, and D. Gil. We would also like to acknowledge the IBM Research Cognitive Compute Cluster and the Center for Computational Innovation at Rensselaer Polytechnic Institute for computational resources on the AiMOS Supercomputer.

## Author contributions

C.M., J.T., and G.W.B. contributed to the development of the weight programming optimisation framework; M.J.R., M.L., S.R.N., P.N., S.A., A.F., J.L., A.F., A.S. helped characterise and model various hardware non-idealities considered during hardware-aware DNN training and inference simulations; A.N. and H.T. contributed the BERT-base simulation results; R.L.B. and N.L. characterised and provided PCM device models; and all authors contributed to writing and editing of the manuscript.

## Competing interests

The authors declare no competing interests.
