## [Peer Review File · Nature Communications]

Title: Optimised Weight Programming for Analogue
Memory-based Deep Neural NetworksREVIEWER COMMENTS

Reviewer #1 (Remarks to the Author):

Key results

This paper presents a weight programming optimisation approach to aid mapping of a pre-trained artificial neural network (ANN) onto an analogue-memory-based ANN accelerator.

The authors aim to show that by following their Differential Weight Evolution (DWE) approach, it is possible to map a pre-trained ANN onto analogue hardware with minimal inference error relative to the original digital implementation of the ANN. The method relies on applying a weight programming optimisation step, independent of inference testing, which simplifies the mapping challenge to a reduced-dimension optimisation problem, which is efficiently solved using a differential evolution algorithm. The method relies on discretising the distribution of weights to be mapped, and minimising the error relative to the trained ANN weights, rather than using an inference-based objective function. The authors demonstrate the algorithm applied to a range of ANN structures (CNN, LSTM, etc), covering a range of applications/datasets. The approach is also applied to two different device characteristics, demonstrating the approach working for more than one device, potentially providing a generic approach to optimal weight programming for analogue-memory-based ANN accelerators.

Validity

The method is tested on a range of DNN architectures, including RNNs and CNNs, demonstrating the general applicability of the approach. This strengthens the generality of the method, as key architectures are covered (LSTM, CNN), on a range of different problems (visual classification, NLP).

The method is also claimed to work on a range of analogue-memory-based ANN accelerators, and the authors quote its applicability to: resistive RAM (ReRAM), conductive-bridging RAM (CBRAM), NOR flash, magnetic RAM (MRAM), and phase-change memory (PCM). While this may well be the case, it is not clear if the data presented actually supports this.

For me, it was hard to understand what type of device was simulated to produce the data in Fig 4. And while Fig 5 contained the description in the caption "An alternative device that with different underlying stochastic analogue memory device models", it wasn't clear how the simulated device differed, or how these different characteristics challenged the weight programming optimisation approach.

It is acknowledged that page 23 contains the sentence: "underlying analogue memory conductance drift model has been modified to a match a different Phase-Change Memory (PCM) device previously reported". However it is felt that at least a general description of the target devices should be included in this work, as it's fundamental to the generality of the findings.

Significance

The presented method and results are of key interest and significance to a broad sector of the

community, as the quest for low power AI hardware has great potential, and is relevant to data scientists, hardware designers, and edge computing.

The work shows a methodical and rigorous approach to solve the new challenge of mapping pre-trained ANN models to analogue-memory-based hardware. This hardware is inherently unpredictable, and therefore presents challenges when targeting a reliable accurate implementation. This work presents a method to overcome these challenges, which is a significant step for device manufacturers, and opens the door to more ambitious future applications.

Data and methodology

The presented approach is described in detail, however there are a large number of constituent parts, which makes covering all aspects a challenge. The method as understood by this reviewer from reading the manuscript and supplementary material follows the steps:

1. Train DNN using standard methods (digital hardware with double precision arithmetic) to find baseline
2. Re-train DNN using hardware-aware training (including device non-idealities)
3. Discretise resulting weight distribution to reduce parameter space
4. Apply DWE optimisation to find optimal conductance programming configuration

It is assumed step 1 follows standard practice, therefore does not require comment.

Step 2 covers hardware-aware (HWA) training, which is demonstrated to be a complementary step to DWE conductance programming. As understood from the text, during training noise was added to weight parameters when calculating the output of the ANN, however this noise was not included when performing training using error backpropagation and SGD (with weight updates performed using full precision). However, once trained, inference is performed on the testing set including a much larger range of device non-idealities, including: MAC cycle-to-cycle non-idealities, PCM programming noise, read noise, $1/f$ noise, conductance-dependent drift, drift variability, and conductance drift.

It is felt the manuscript is slightly misleading around the definitions of HWA training, and would benefit from clarity around exactly what is included, and where. For example, the first sentence of the methods reads: "...We incorporate hardware-specific non-idealities during forward...", however if I've understood correctly, only one non-ideality is included (weight noise). This observation is not meant as a criticism, and in fact could be viewed as a positive of the DWE method, in that it is able to succeed without considering additional device characteristics. However, an important aspect of future work in this area could be how to integrate further device non-idealities into training and programming, and clarification in this work would help the reader understand the interplay between the two steps.

Step 3 takes the HWA trained network and discretises the resulting weight distribution, reducing the parameter space of the subsequent weight programming optimisation problem. While this represents a pragmatic and logical approach, the method would benefit from additional discussion on how the discretised weights for optimal fitting are selected. For example, the text states "The reported results

make use of six discretized weight intervals”, however it is not discussed how this number is selected, or whether the intervals are all equal. Presumably this discretisation will have an effect on overall system performance, along with the weight programming optimisation. Additionally, would different weight distributions (achieved from HWA training) benefit from different discretisations?

Step 4 performs the weight programming optimisation, using the standard Differential Evolution algorithm from SciPy. A number of additional algorithms are discussed for the optimisation process, and the challenges around gradient-based methods are covered well (although no data is presented). A key feature of the optimisation process is the removal of inference from this step, instead opting for an objective function capturing the difference between the HWA trained weights and those programmed via the hardware simulator. A time-averaged, normalised mean-squared-error metric is introduced as a proxy for inference accuracy, with all weight errors treated as equal, regardless of weight value (as no correlation between weight magnitude/sign was observed with partial gradient of loss function at last epoch of training). While a relatively straightforward approach, this technique reduces optimisation time, and enables modularisation of the training and weight programming steps.

Analytical approach

The results obtained appear suitable for the field, with mean/standard deviations reported for testing where appropriate. The breadth of testing is also a strong attribute of the work, with multiple different ANNs tested, demonstrating application of the proposed techniques over a range of scenarios.

The supplementary material reinforces the data of the main paper, with additional results from training the baseline network and exploring the effect of weight noise on system performance (as discussed above, it is understood that this stochasticity in the forward pass constitutes HWA training in this work).

Suggested improvements

Overall the manuscript is produced to a high standard, and the work makes a solid contribution to the field.

To improve the work it is suggested to:

- Clarify the description of hardware aware training, highlighting explicitly which hardware non-idealities were included in the work.
- Clarify how the results were produced – i.e. that the results were produced using simulated hardware, rather than physical devices
- Clarify the characteristics of the different hardware devices targeted in Figures 4 and 5, and describe how covering these two devices represents general coverage of all analogue-memory-based ANN accelerators.
- Add further discussion around the weight discretisation process

A few minor points which would also benefit clarification:

- Definition of alpha is confusing on page 9: ‘digital scale factor’ or ‘drift compensation factor’
- ‘simple’ -> ‘simply’: page 12, 2nd paragraph.

- Clarify whether ‘floating-point’ refers to 32 or 64-bit representations
- Gamma is used as learning rate decay factor in the supplementary material, but as the relative importance of the discretised weight (on page 14) – suggest to stick to one definition per symbol.
- In Figure 4 and Figure 5, it was unclear to me what exactly was meant by the labels: MSP only; MSP/LSP 50:50; Optimized. Does this refer to $F=1$ in the 50:50 case, and $F=2$ for the Optimized case?

Clarity and context

The manuscript provides an excellent introduction to the topic of analogue-memory-based ANN hardware, and the challenges around programming it accurately and reliably.

The text is also positioned well for a general reader, increasing the accessibility of the work to a wide audience.

References

References appear appropriate for the presented work.

Your expertise

As an academic researcher in the field of neuromorphic computing, my experience lies in bio-inspired algorithms and their mapping to novel hardware devices and systems.

Reviewer #2 (Remarks to the Author):

Analog in-memory computing based NN research has attracted quite a lot of attentions recently due to its unmatched energy efficiency. However, one of the major concerns is the accuracy loss caused by all kinds of nonideality factors in the analog domain. This paper presented a weight optimization scheme for PCM based analog in-memory computing, which can greatly improve the adaptability of the analog neural networks. Their proposed method was developed based on a PCM pair structure, which has been published and studied in several papers before by the same group. In this paper, the authors claimed that their proposed solution is agnostic to DNN structure and has a good compatibility with different memory structures. Although the overall approach and results appear to be promising, the technical analysis included in the paper is not explained in detail. It would be better if the authors could put more related technical illustrations about the hardware aware training algorithm and the hyper parameter scan method in the paper. The material in the supplementary section did not provide enough information to support the method used in the main text.

A few comments and suggestions:

1. Conductance $G+$, $G-$, $g+$, $g-$ are not clearly defined at the first place where they appeared. It would be great if the author could add a figure to show the detailed circuit implementation of one weight and illustrate the connection between circuit schematic and those conductance pairs.
2. Not 100% sure about the circuit and weight structure. Are $G+$ and $g+$ used for the positive weight,

while G- and g- are in charge of the negative weight?

3. What is the difference between MSP and LSP? Is there any physical difference between those two pairs, or just the programmed conductance values are different? How the factor F was implemented? Physical size ratio? This factor was choosing to be 4 in several previous published papers. Why switched to 2 in this paper?

4. MSP and LSP can be misleading in the paper, as there are only two pairs used to compose the actual weight. "Most significant" and "least significant" are normally used in the case where a string of elements are sorted in a row, like MSB and LSB. For the scenario in this paper, coarse tuning pair and fine tuning pair could be a better option.

5. For the distribution plots on the background of Fig. 2 (b) and (e), I did not find any corresponding descriptions in the main text nor the caption talking about them. People may get confused when looking at those figures.

6. The granularity of the weight in Fig. 2 (b) (e), Fig. 4 (g-i), and Fig. 5 (g-i) is huge. What are the physical limitations or other considerations for pursuing a smaller conductance resolution? This resolution is directly related to the size of the hyper dimensional parameter space. So, what will be the consequence if we want to have a finer resolution?

7. On page 28, the authors state that all the hardware non-idealities, noise, $1/f$ noise, etc., have been considered in their proposed approach. However, it is not very clear to me how those factors were modeled and considered? Are they considered separately or as a whole? What if we need to take something else into account, for instance, the nonlinearity of the data converters?

Reviewer #3 (Remarks to the Author):

The manuscript by Mackin et al. describes an extensive methodology to optimise programming for unideal weights that are based on analogue hardware devices such as PCM or RRAM. The methodology is generalised for all neural network structures and is modeled using an effective computation technique. The concept is highly innovative and treats an issue or challenge that is hindering wide-spread application of hardware-based neural networks at the moment, and is preventing those systems from superiority of traditional software-based systems.

For this reviewer, the main issue with this work lies with the presentation of the manuscript, which is sometimes too hard to read and follow. Although the figures and separate sections seem clear to follow, the complete story is sometimes lost on me. There is a lot of jargon used, and many abbreviations. Some of the figures are also hard to fully grasp, as the authors wanted to capture a lot of information in one graph.

Perhaps it is my materials background that makes it more difficult for me to understand, but I believe that the work could be improved if the story gets somewhat more clearer and to the point, and some graphs are explained in more layman's terms. Overall I would certainly advise publication in Nature Communications.

A few comments:

1. On page 8 the authors introduce the factor F . This factor seems to be of significance for the results (page 24) but it is not completely clear how this factor is chosen.
2. The method to reduce computational load is to see if the weight error minimisation should improve DNN accuracy. I might be misunderstanding but how can one distinguish this from actual DNN improvement by training over time? Wouldn't you need to have already trained a DNN to the maximum accuracy for that, and thus still need a lot of computational load?
3. Figure 3f is not completely clear to me what it is representing and how to read the optimum weight programming strategy.
4. Figure 3g is difficult to read because of the light shades of blue
5. Page 18 ("Although the constrained [...] for many optimizers") is not clear to me. Can the authors further explain this, perhaps with an example?

REVIEWER COMMENTS

Reviewer #1 (Remarks to the Author):

Key results

This paper presents a weight programming optimisation approach to aid mapping of a pre-trained artificial neural network (ANN) onto an analogue-memory-based ANN accelerator.

The authors aim to show that by following their Differential Weight Evolution (DWE) approach, it is possible to map a pre-trained ANN onto analogue hardware with minimal inference error relative to the original digital implementation of the ANN. The method relies on applying a weight programming optimisation step, independent of inference testing, which simplifies the mapping challenge to a reduced-dimension optimisation problem, which is efficiently solved using a differential evolution algorithm. The method relies on discretising the distribution of weights to be mapped, and minimising the error relative to the trained ANN weights, rather than using an inference-based objective function. The authors demonstrate the realgorithm applied to a range of ANN structures (CNN, LSTM, etc), covering a range of applications/datasets. The approach is also applied to two different device characteristics, demonstrating the approach working for more than one device, potentially providing a generic approach to optimal weight programming for analogue-memory-based ANN accelerators.

Validity

The method is tested on a range of DNN architectures, including RNNs and CNNs, demonstrating the general applicability of the approach. This strengthens the generality of the method, as key architectures are covered (LSTM, CNN), on a range of different problems (visual classification, NLP).

The method is also claimed to work on a range of analogue-memory-based ANN accelerators, and the authors quote its applicability to: resistive RAM (ReRAM), conductive-bridging RAM (CBRAM), NOR flash, magnetic RAM (MRAM), and phase-change memory (PCM). While this may well be the case, it is not clear if the data presented actually supports this.

For me, it was hard to understand what type of device was simulated to produce the data in Fig 4. And while Fig 5 contained the description in the caption "An alternative device that with different underlying stochastic analogue memory device models", it wasn't clear how the simulated device differed, or how these different characteristics challenged the weight programming optimisation approach.

We agree with the reviewer's comment that it was not clear from the caption what type of device the various models and results were derived from. We have added the following general description of the memory device models at the beginning of the section:

Analogue memory device models are experimentally derived from mushroom-type phase change memory (PCM) devices. Figure 4a-c depicts the set of stochastic device models that define programming errors, drift coefficients, and read noise, all as a function of conductance. For each device characteristic, both the average response (solid line) and the variability around this (shaded region, corresponding to plus-minus one standard deviation) are used by the optimization algorithm.

To bring further clarity, we have also revised the Figure 4 caption to reflect this.

The degree of characterization presented in this work for Phase-Change Memory (PCM) is not readily available for other types of analogue memory. As a result, we cannot run weight programming optimization simulations with the same level of rigor for other analogue memories at this time. We did, however, run simulations for a variety of PCM devices with different characteristics (in the Supplementary Information) to show the optimization framework consistently improved inference accuracy over time. This includes less intuitive strategies where non-zero values are programmed in g^- conductances even for positive weights (Figure 4h), which is now pointed out in the main text. Figure 3f also shows that the optimiser has good coverage over the range of valid weight programming strategies.

Given the very high dimensionality of the DNN, there is virtually zero chance that our optimized weight programming strategies are coincidentally stumbling into better (and more temporally stable) minima consistently. Because of this, we are confident that given the generality of the optimization framework and its high degree of flexibility, that it can be used other types of analogue memory, which also exhibit programming errors, read noise, and potentially drift—just with different characteristics. We have, however, revised statements regarding applicability to other analogue memories to the following:

Being a highly flexible numerical heuristic, our approach can accommodate arbitrary device-level complexity, making it potentially relevant for a variety of analogue memories and their continually evolving device characteristics.

We have also included a brief section in the Supplementary Information to provide some detail on the challenge posed by this problem for the optimizer.

It is acknowledged that page 23 contains the sentence: “underlying analogue memory conductance drift model has been modified to a match a different Phase-Change Memory (PCM) device previously reported”. However it is felt that at least a general description of the target devices should be included in this work, as it’s fundamental to the generality of the findings.

We have now included a general description of our PCM devices in the main text as well as Methods section on PCM Characterization.

The description of our memory devices was also moved to the beginning of the section to improve flow and to textually introduce the elements of Figure 4 in the same order that they appear graphically.

Significance

The presented method and results are of key interest and significance to a broad sector of the community, as the quest for low power AI hardware has great potential, and is relevant to data scientists, hardware designers, and edge computing.

The work shows a methodical and rigorous approach to solve the new challenge of mapping pre-trained ANN models to analogue-memory-based hardware. This hardware is inherently unpredictable, and therefore presents challenges when targeting a reliable accurate implementation. This work presents a method to overcome these challenges, which is a significant step for device manufacturers, and opens the door to more ambitious future applications.

Data and methodology

The presented approach is described in detail, however there are a large number of constituent parts, which makes covering all aspects a challenge. The method as understood by this reviewer from reading the manuscript and supplementary material follows the steps:

1. Train DNN using standard methods (digital hardware with double precision arithmetic) to find baseline
2. Re-train DNN using hardware-aware training (including device non-idealities)
3. Discretise resulting weight distribution to reduce parameter space
4. Apply DWE optimisation to find optimal conductance programming configuration

It is assumed step 1 follows standard practice, therefore does not require comment.

Step 2 covers hardware-aware (HWA) training, which is demonstrated to be a complementary step to DWE conductance programming. As understood from the text, during training noise was added to weight parameters when calculating the output of the ANN, however this noise was not included when performing training using error backpropagation and SGD (with weight updates performed using full precision). However, once trained, inference is performed on the testing set including a much larger range of device non-idealities, including: MAC cycle-to-cycle non-idealities, PCM programming noise, read noise, $1/f$ noise, conductance-dependent drift, drift variability, and conductance drift.

It is felt the manuscript is slightly misleading around the definitions of HWA training, and would benefit from clarity around exactly what is included, and where. For example, the first sentence of the methods reads: "...We incorporate hardware-specific non-idealities during forward...", however if I've understood correctly, only one non-ideality is included (weight noise). This observation is not meant as a criticism, and in fact could be viewed as a positive of the DWE method, in that it is able to succeed without considering additional device characteristics. However, an important aspect of future work in this area could be how to integrate further device non-idealities into training and programming, and clarification in this work would help the reader understand the interplay between the two steps.

We apologize for the misunderstanding here. The hardware-aware training was performed with several non-idealities present (e.g. weight noise, activation discretization due to ADCs and PWMs, etc.) during the forward pass. Conductance drift is not included because its time-dependence is not readily incorporated into the loss function during training. Similarly, the use of multiple conductances per weight is not included because optimal weight splitting, which also exhibits time-dependence, cannot readily be incorporated into the backpropagation algorithm and loss function during training. Weight updates during backpropagation, however, are ideal and performed at full precision (i.e. without any hardware non-idealities). We have revised the description of the hardware-aware training under Methods, to more accurately reflect this.

We have also revised a sentence on page 21, which may have been the source of some confusion here. It previously only mentioned applying weight noise during training, which was an inaccurate statement. This sentence has been revised to the following:

By subjecting the DNN to memory and circuit non-idealities during training, HWA-training clearly makes the DNN more resilient to the various hardware non-idealities,

and significantly enhances network accuracy relative to floating-point training across the board.

Step 3 takes the HWA trained network and discretises the resulting weight distribution, reducing the parameter space of the subsequent weight programming optimisation problem. While this represents a pragmatic and logical approach, the method would benefit from additional discussion on how the discretised weights for optimal fitting are selected. For example, the text states “The reported results make use of six discretized weight intervals”, however it is not discussed how this number is selected, or whether the intervals are all equal. Presumably this discretisation will have an effect on overall system performance, along with the weight programming optimisation. Additionally, would different weight distributions (achieved from HWA training) benefit from different discretisations?

This is an excellent point, and we thank you for bringing attention to this. The manuscript has been revised to state that the discretized points of the weight programming optimization strategy are equally spaced. In addition, we added the following text to the manuscript to provides some clarity as to how this number was chosen:

We chose six points as a compromise between capturing the complexity of the optimized weight programming strategy and compute time (as each additional discretized point adds four dimensions to the search space). It is important to note that different memory device characteristics and network weight distributions could lead to instances where having more (or fewer) discretization points is beneficial. Additional details on how the number of points D impacts the optimized weight programming strategy are provided in the Supplementary Information.

We have also added a section to the Supplementary Information that shows how the number of discretization points impacts the resulting optimal weight programming strategy and inference accuracy performance.

Step 4 performs the weight programming optimisation, using the standard Differential Evolution algorithm from SciPy. A number of additional algorithms are discussed for the optimisation process, and the challenges around gradient-based methods are covered well (although no data is presented). A key feature of the optimisation process is the removal of inference from this step, instead opting for an objective function capturing the difference between the HWA trained weights and those programmed via the hardware simulator. A time-averaged, normalised mean-squared-error metric is introduced as a proxy for inference accuracy, with all weight errors treated as equal, regardless of weight value (as no correlation between weight magnitude/sign was observed with partial gradient of loss function at last epoch of training). While a relatively straightforward approach, this technique reduces optimisation time, and enables modularisation of the training and weight programming steps.

We did not feel it was necessary to include data from other optimizers as they completely failed in minimizing the objective function, so they could not provide inference accuracy improvements over the MSP only and F=1 MPS/LSP (50/50) programming strategies.

We now explicitly mention that this is a non-convex problem in the main text because the numerically evaluated Hessian of the error metric is not positive semi-definite. This poses a challenge for gradient descent-based methods. We also include a section in the

Supplementary Information to further illustrate the challenge posed by this problem for the optimizer.

Analytical approach

The results obtained appear suitable for the field, with mean/standard deviations reported for testing where appropriate. The breadth of testing is also a strong attribute of the work, with multiple different ANNs tested, demonstrating application of the proposed techniques over a range of scenarios.

The supplementary material reinforces the data of the main paper, with additional results from training the baseline network and exploring the effect of weight noise on system performance (as discussed above, it is understood that this stochasticity in the forward pass constitutes HWA training in this work).

Suggested improvements

Overall the manuscript is produced to a high standard, and the work makes a solid contribution to the field.

To improve the work it is suggested to:

- Clarify the description of hardware aware training, highlighting explicitly which hardware non-idealities were included in the work.

We have revised the hardware-aware training description under the Methods section to explicitly highlight which hardware non-idealities are included in this work.

For clarity, we would like to emphasize the point of the paper, however, is not the hardware-aware training, but the optimal translation software trained DNNs (hardware-aware or otherwise) into analogue memory-based DNN accelerators. This is evidenced by the fact that we perform 32-bit floating-point training and hardware-aware training and in both cases optimized weight programming exists independently of the training method and improves inference accuracy.

We have also included additional details regarding the hardware-aware training and hyperparameter scans in the Supplementary Information.

- Clarify how the results were produced – i.e. that the results were produced using simulated hardware, rather than physical devices

We thank the reviewer for bringing this to our attention. The manuscript has been revised to explicitly state that “all reported inference results are produced using simulated hardware as opposed to physical hardware.” In addition, the captions of Figures 4 and 5 have been revised to explicitly state that these are simulated inference results.

- Clarify the characteristics of the different hardware devices targeted in Figures 4 and 5, and describe how covering these two devices represents general coverage of all analogue-memory-based ANN accelerators.

The reviewer makes an excellent point here. It is impossible to prove the existence of, and the ability of our optimization framework to find, improved weight programming schemes for all forms of analogue memory, across all potential device characteristics.

The degree of characterization presented in this work for Phase-Change Memory (PCM) is not readily available for other types of analogue memory. We did, however, run simulations for a variety of different PCM characteristics (in the Supplementary Information) to show that the optimization framework consistently improved inference accuracy over time. In addition, we have deliberately made this framework modular and flexible so that additional memory nonidealities and other considerations such as algorithmic drift compensation (Figure 1 of the main text), can be readily incorporated if necessary.

- Add further discussion around the weight discretisation process

Please see the previous comment addressing the number of discretization points.

A few minor points which would also benefit clarification:

- Definition of alpha is confusing on page 9: 'digital scale factor' or 'drift compensation factor'

We thank the reviewer for noticing this. The manuscript has been revised to only refer to alpha as a drift compensation factor for consistency.

- 'simple' -> 'simply': page 12, 2nd paragraph.

This has been corrected. Thank you.

- Clarify whether 'floating-point' refers to 32 or 64-bit representations

Floating-point has been revised to 32-bit floating-point for clarity throughout the text. We have also revised the Figure 4 caption to specifically mention that FP denotes 32-bit floating-point as well.

- Gamma is used as learning rate decay factor in the supplementary material, but as the relative importance of the discretised weight (on page 14) – suggest to stick to one definition per symbol.

We thank the reviewer for noticing this. We have changed the relative importance of the discretized weight from gamma to kappa throughout the manuscript. The learning rate decay factor presented in the supplementary material remains as gamma.

- In Figure 4 and Figure 5, it was unclear to me what exactly was meant by the labels: MSP only; MSP/LSP 50:50; Optimized. Does this refer to $F=1$ in the 50:50 case, and $F=2$ for the Optimized case?

We thank the reviewer for bringing attention to the potential ambiguity regarding these programming strategies. "MSP only" means the majority (virtually all) of the weight is programmed in the MSP with the LSP only used for minimal tuning. This is mentioned in the results section with a specific reference back to the sub-optimal programming strategy highlighted in Figure 2b. The MSP/LSP 50:50 is meant to be $F=1$, MSP/LSP (50:50). The text has been revised to explicitly state that the F factor is 1. It also provides a reference to bit slicing approach in which each slice has equal significance.

Clarity and context

The manuscript provides an excellent introduction to the topic of analogue-memory-based ANN hardware, and the challenges around programming it accurately and reliably.

The text is also positioned well for a general reader, increasing the accessibility of the work to a wide audience.

References

References appear appropriate for the presented work.

Your expertise

As an academic researcher in the field of neuromorphic computing, my experience lies in bio-inspired algorithms and their mapping to novel hardware devices and systems.

Reviewer #2 (Remarks to the Author):

Analog in-memory computing based NN research has attracted quite a lot of attentions recently due to its unmatched energy efficiency. However, one of the major concerns is the accuracy loss caused by all kinds of nonideality factors in the analog domain. This paper presented a weight optimization scheme for PCM based analog in-memory computing, which can greatly improve the adaptability of the analog neural networks. Their proposed method was developed based on a PCM pair structure, which has been published and studied in several papers before by the same group. In this paper, the authors claimed that their proposed solution is agnostic to DNN structure and has a good compatibility with different memory structures. Although the overall approach and results appear to be promising, the technical analysis included in the paper is not explained in detail. It would be better if the authors could put more related technical illustrations about the hardware aware training algorithm and the hyper parameter scan method in the paper. The material in the supplementary section did not provide enough information to support the method used in the main text.

We thank reviewer #2 for the feedback. We did not focus heavily on the specifics of our hardware-aware training because weight programming optimization exists as a largely independent step for translating DNNs into analogue hardware, regardless of how that particular DNN was obtained (through hardware-aware training or otherwise). This is shown by the improvement in inference accuracy even for non-hardware-aware trained networks, such as the FP-trained LSTM (Figure 4). We included the following statement in the text to further emphasize this:

It should also be pointed out that weight programming optimization represents a critical step in translating DNNs into analogue hardware, irrespective of how those DNNs were originally obtained—through hardware-aware training or otherwise. We focus primarily on hardware-aware trained DNNs in this work to show that weight programming optimization can extend and *augment* the inference accuracy benefits of hardware-aware training. These largely independent steps, when combined, help analogue memory-based DNNs reach and maintain inference accuracies that are equivalent (or near equivalent) to 32-bit floating-point trained DNNs.

With that said, we have also provided more details on the specifics of our hardware-aware training in the main text and Supplementary Information. More detailed descriptions of the specific hardware-aware training techniques used can also be found in the following references, cited in this paper:

[19] Joshi, V. et al. Accurate deep neural network inference using computational phase-change memory. *Nature Communications* 11, 1–13 (2020).

[22] Rasch, M. et al. A flexible and fast pytorch toolkit for simulating training and inference on analog crossbar arrays. *IEEE International Conference on Artificial Intelligence Circuits and Systems (AICAS)* 1–4 (2021).

A few comments and suggestions:

1. Conductance G^+ , G^- , g^+ , g^- are not clearly defined at the first place where they appeared. It would be great if the author could add a figure to show the detailed circuit implementation of one weight and illustrate the connection between circuit schematic and those conductance pairs.

We thank the reviewer for the comments. We state in the caption of Figure 1 that G^+ , G^- , g^+ and g^- represent analogue memory conductance values. There are many ways to implement a synaptic weight as described by $W = F(G^+ - G^-) + (g^+ - g^-)$ from a circuit standpoint. We left out the circuit specifics here to emphasize that the issue of how to optimally translate weights into analogue hardware exists regardless of how the unit cell corresponding to each synaptic weight might be implemented in circuitry. Similarly, we do not initially introduce the analogue memory as PCM to emphasize the fact that this fundamental issue that exists regardless of analogue memory type.

We have added two references in the main text to provide some additional detail on specific circuit implementation examples:

[28] Chang, H.-Y. et al. Ai hardware acceleration with analog memory: Microarchitectures for low energy at high speed. *IBM Journal of Research and Development* 63, 8:1–8:14 (2019).

[29] Narayanan, P. et al. Fully on-chip mac at 14nm enabled by accurate row-wise programming of pcm-based weights and parallel vector-transport in duration-format. In *2021 Symposium on VLSI Technology*, 1–2 (2021).

2. Not 100% sure about the circuit and weight structure. Are G^+ and g^+ used for the positive weight, while G^- and g^- are in charge of the negative weight?

G^+ and g^+ make positive contributions to the overall weight value while G^- and g^- make negative contributions, where each synaptic weight is implemented as $W = F(G^+ - G^-) + (g^+ - g^-)$. That is not to say, however, that a positive weight cannot have some non-zero value in G^- and g^- . Similarly, a negative weight can also have some non-zero value stored in the G^+ and g^+ conductances. That is precisely the point of this paper. There may exist some

less intuitive programming strategies where we may wish to program non-zero values into the “negative” conductances despite the overall weight being positive because it moves us into a more favorable conductance range (either based on the associated programming errors, drift rates, or read noise). This is seen in Figure 4h, for instance, where the optimal weight programming strategy programs non-zero values into g^- despite the overall weight being positive. We have now added the following sentence to the text to provide some added clarity and stress this important point:

It is interesting to note that Figure 4h finds an optimal weight programming strategy that programs non-zero values into the g^- conductance even when the overall weight being implemented is positive. It is precisely these complex and potentially counter-intuitive programming strategies, that also translate into improved inference accuracy, that our optimization framework allows us to find.

3. What is the difference between MSP and LSP? Is there any physical difference between those two pairs, or just the programmed conductance values are different? How the factor F was implemented? Physical size ratio? This factor was choosing to be 4 in several previous published papers. Why switched to 2 in this paper?

There is no physical difference. All devices are fabricated identically. We have now explicitly stated this in manuscript for clarity:

Conductances within a weight are identically fabricated but may also have varying significance as determined by the scaling factor F between the Most Significant Pair (MSP) and Least Significant Pair (LSP).

The F factor can be implemented in several different ways. In the case where analogue activations are implemented as pulse durations, we can simply send a F times longer pulse duration to the MSP relative to LSP. Another simple way to implement the F factor is to use current mirrors with a current gain factor of F on the MSP relative to the LSP current contribution. We added two references (mentioned above) to provide a general idea on how input activations might be encoded, although there are many different ways to implement F factor in peripheral circuits.

In general, we have found F factors in the range of 1-5 to be advantageous. In a previous paper [27], we found that $F=4$ optimized the weight programming for a particular type of PCM device. In another paper [4], we found that an $F=3$ worked best. This is obviously dependent on the device characteristics. Also, neither of the previous papers optimized weight programming to the extent that was done here. For instance, [27] only focused on minimizing programming errors at $t=0$. This work solves a much more complex problem of optimizing weight programming over multiple time steps in the presence of programming errors, drift, and read noise (each with conductance dependence). This work also allows for much more complex weight programming strategies in that weights distributed across multiple conductances can vary as a function of the target, i.e. $G^+(W)$, $G^-(W)$, $g^+(W)$, and $g^-(W)$.

[27] Mackin, et al., “Weight Programming Optimization in DNN Hardware Accelerators in Presence of NVM Variability,” *Advanced Electronic Materials*, 2019.

[4] Ambrogio, et al., “Equivalent-accuracy accelerated neural-network training using analogue memory,” *Nature*, 2018.

4. MSP and LSP can be misleading in the paper, as there are only two pairs used to compose the actual weight. “Most significant” and “least significant” are normally used in the case where a string of elements are sorted in a row, like MSB and LSB. For the scenario in this paper, coarse tuning pair and fine tuning pair could be a better option.

We agree with the reviewer that MSP and LSP typically refer to more than two pairs of conductances used to comprise a single weight. We would like to maintain the MSP and LSP terminology for consistency with our previously published works.

5. For the distribution plots on the background of Fig. 2 (b) and (e), I did not find any corresponding descriptions in the main text nor the caption talking about them. People may get confused when looking at those figures.

We thank the reviewer for bringing attention to this. We added a sentence to the main text (page 9) describing that this background distribution represents the target weight distribution.

The weight distribution being programmed is shown in the background (blue).

We also updated the captions of Figure 2 and Figure 4 to make this clearer.

6. The granularity of the weight in Fig. 2 (b) (e), Fig. 4 (g-i), and Fig. 5 (g-i) is huge. What are the physical limitations or other considerations for pursuing a smaller conductance resolution? This resolution is directly related to the size of the hyper dimensional parameter space. So, what will be the consequence if we want to have a finer resolution?

The reviewer is correct that granularity here comes at the expense of increased computational expense, where each additional discretization point introduces another four dimensions to the overall search space. This question is similar to a concern from Reviewer #1, which we addressed by adding some clarifying text to the manuscript and by adding a section in the Supplementary Information (see comment above).

7. On page 28, the authors state that all the hardware non-idealities, noise, $1/f$ noise, etc., have been considered in their proposed approach. However, it is not very clear to me how those factors were modeled and considered? Are they considered separately or as a whole? What if we need to take something else into account, for instance, the nonlinearity of the data converters?

We thank the reviewer for bringing attention to this. Hardware nonidealities are considered as a whole during training (with the exception of conductance drift and multi-PCM weight implementations, which cannot readily be incorporated into the training phase). We have revised the text considerably to make this clearer. We have also expanded the methods section to include more details regarding our hardware-aware training.

The nonlinearity of the data-converters and any other hardware associated imperfections should be captured during the hardware-aware training phase. This is independent of the subsequent weight programming optimization, which governs how to best translate trained

weights into the hardware. We have also added some text to the conclusion and Methods sections to further clarify this.

Reviewer #3 (Remarks to the Author):

The manuscript by Mackin et al. describes an extensive methodology to optimise programming for unideal weights that are based on analogue hardware devices such as PCM or RRAM. The methodology is generalised for all neural network structures and is modeled using an effective computation technique. The concept is highly innovative and treats an issue or challenge that is hindering wide-spread application of hardware-based neural networks at the moment, and is preventing those systems from superiority of traditional software-based systems.

For this reviewer, the main issue with this work lies with the presentation of the manuscript, which is sometimes too hard to read and follow. Although the figures and separate sections seem clear to follow, the complete story is sometimes lost on me. There is a lot of jargon used, and many abbreviations. Some of the figures are also hard to fully grasp, as the authors wanted to capture a lot of information in one graph.

Perhaps it is my materials background that makes it more difficult for me to understand, but I believe that the work could be improved if the story gets somewhat more clearer and to the point, and some graphs are explained in more layman's terms. Overall I would certainly advise publication in Nature Communications.

We thank the reviewer for their feedback. We have made several revisions to hopefully bring added clarity. We have also restructured the text in some places so that content is introduced the same order that it appears graphically in the figures.

A few comments:

1. On page 8 the authors introduce the factor F . This factor seems to be of significance for the results (page 24) but it is not completely clear how this factor is chosen.

The MSP/LSP scale factor F is also part of the optimization process. This is described on page 11.

We have also modified the caption of Figure 4 on (page 24) to reiterate this point.

2. The method to reduce computational load is to see if the weight error minimisation should improve DNN accuracy. I might be misunderstanding but how can one distinguish this from actual DNN improvement by training over time? Wouldn't you need to have already trained a DNN to the maximum accuracy for that, and thus still need a lot of computational load?

Correct, we reduce computational load of the weight programming optimization problem by using a less computationally expensive error metric as a proxy for inference accuracy. We then show that minimizing this error metric indeed has the effect finding weight programming strategies that improve inference accuracy, including over time.

If I understand the second question correctly, there is no straightforward way to incorporate the time dependent nature of drift during DNN training. Additionally, it is not apparent how one might modify the backpropagation algorithm so that weight updates are split optimally across multiple conductances (since this is how weights are implemented). As a result, we have broken the DNN inference accuracy optimization process into two steps. The first step is to perform hardware-aware training to enhance the overall resilience of the network to circuit and memory imperfections. This typically results in single-valued and unitless weights. The second step is to optimize the translation of these hardware-aware trained weights into analogue memory, where each weight is now in units of microSiemens and comprised of multiple conductances.

The maximum achievable DNN accuracy is computed using 32-bit floating-point (shown in Figure 4, 5 as the dash-dot purple baseline). Our goal is to push our analogue memory-based DNNs as close as possible to this ideal baseline, despite their many nonidealities. In this way, we realize the throughput and energy efficiency benefits of analogue memory-based DNNs while minimizing potential tradeoffs in inference accuracy.

3. Figure 3f is not completely clear to me what it is representing and how to read the optimum weight programming strategy.

We thank the reviewer for bringing attention to this and have revised the Figure 3 caption to state:

f) A two-dimensional projection of the weight programming strategies, including the optimal solution (solid lines). Background violin plots show coverage of the weight programming space explored and reveal underlying programming constraints.

The optimal weight programming strategy describes how to best map unitless software weights into hardware. The optimal programming strategy is described by the functions $G^+(W)$, $G^-(W)$, $g^+(W)$, and $g^-(W)$, which are depicted in Figure 3f. Here W is the unitless software weight we wish to implement, and each function simply returns the appropriate conductance that should be programmed in hardware when implementing that weight.

We also note a description of Figure 3f at the top of page 16, which provides some additional detail on the interpretation of the background violin plots and their significance.

4. Figure 3g is difficult to read because of the light shades of blue

We thank the reviewer for the feedback. We have darkened the light shades of blue in Figure 3g so that it is easier to read.

5. Page 18 (“Although the constrained [...] for many optimizers”) is not clear to me. Can the authors further explain this, perhaps with an example?

The conductance parameter constraints are inter-dependent and can be quite complex. This is partially captured in Figure 3f. Here small weights with a value of $0 \mu\text{S}$ can be constructed in many different ways as indicated by the heavily overlapping background violin plots. For

instance, all the conductances could be zero. Alternatively, all the conductances could be set to some equal value. The conductances could also be set to a number of other values as long as they cause the weight $W = F(G^+ - G^-) + (g^+ - g^-)$ to sum to zero.

Large weights are more heavily constrained in how they can be implemented (also indicated by the violin plots). In the other extreme, let's say $F=2$ and the maximum conductance we can program is $25 \mu\text{S}$. If we wish to program a weight value of $75 \mu\text{S}$ we must program G^+ and g^+ to the maximum value of $25 \mu\text{S}$. This necessitates programming G^- and g^- to $0 \mu\text{S}$. In this case, the constraints are so tight that there is essentially only one way to implement this weight. Incorporating these constraints also reduces computational expense in that we do not waste time evaluating conductance combinations that do not sum anywhere near the target weight. Generally speaking, however, these conductance constraints are complex and inter-dependent (the G^- constraints depend on the G^+ value, g^+ constraints depend on the G^- value, and the g^- constraints depend on the g^+ value).

To eliminate the optimizer from seeing these complex inter-dependent constraints, we instead optimize parameters from a hypercube, where each parameter has simple and independent constraints from $(0, 1)$. We introduce an intermediate de-normalization step that takes these hypercube parameters and translates them into valid conductance combinations that impose the necessary inter-dependent constraints (previously described). The optimizer now is essentially massaging the parameters in this hypercube while trying to minimize our error metric. Once this optimization process is complete, we apply the same de-normalization step to the optimized hypercube parameters to arrive at the optimal conductance values used in our weight programming strategies.

We have revised the main text (page 17) to add some clarity as to why this represents a high-dimensional non-convex stochastic optimization problem. We have also added a brief section to the Supplementary Information further detailing the difficulty in minimizing the error metric (i.e. improving inference accuracy) beyond the simple MSP/LSP (50/50) weight programming strategy, which serves as our baseline.

REVIEWERS' COMMENTS

Reviewer #1 (Remarks to the Author):

Many thanks to the authors for incorporating review feedback - all questions/queries have been answered/resolved.

Reviewer #2 (Remarks to the Author):

Thanks for the response. The authors have answered most of my questions and concerns listed in the previous review cycle. I am satisfied with their response and the revised manuscript. The overall novelty and analytical quality of the paper have met the criteria for publication.

There is only one minor comment about the revised manuscript:

Those plots in Figure 4 (d, e, f) and Figure 5 (d, e, f) are heavily blocked by the figure legends. Although they are translucent, it is hard to read the data in the covered region. Please consider changing the plotting style of the figures.

Reviewer #3 (Remarks to the Author):

The authors have successfully addressed all my comments and the manuscript is ready for publication.

REVIEWERS' COMMENTS

Reviewer #1 (Remarks to the Author):

Many thanks to the authors for incorporating review feedback - all questions/queries have been answered/resolved.

Reviewer #2 (Remarks to the Author):

Thanks for the response. The authors have answered most of my questions and concerns listed in the previous review cycle. I am satisfied with their response and the revised manuscript. The overall novelty and analytical quality of the paper have met the criteria for publication.

There is only one minor comment about the revised manuscript:

Those plots in Figure 4 (d, e, f) and Figure 5 (d, e, f) are heavily blocked by the figure legends. Although they are translucent, it is hard to read the data in the covered region. Please consider changing the plotting style of the figures.

Thank you for the recommendation. We have revised Figure 4 (d, e, f) and Figure 5 (d, e, f) so less data is covered by the legends.

Reviewer #3 (Remarks to the Author):

The authors have successfully addressed all my comments and the manuscript is ready for publication.